# Co-evolving infectivity and expression patterns drive the diversification of endogenous retroviruses

Kirsten-André Senti [ID][1,✉], Baptiste Rafanel[1,2], Dominik Handler [ID][1], Carolin Kosiol [ID][3], Christian Schlötterer [ID][4] & Julius Brennecke [ID][1,✉]

## Abstract

**Transposable elements are major contributors to the evolution of their hosts, but the mechanisms driving their own diversification remain poorly understood. Here, we reveal key principles governing the evolution of insect endogenous retroviruses (ERVs), a class of infectious LTR-retrotransposons that encode an Envelope protein. Through comparative analyses and experimental studies of transposon replication cycles in *Drosophila*, we demonstrate how two crucial ERV traits—infectivity and spatiotemporal expression—co-evolve. We find that ERVs have adapted their *cis*-regulatory sequences to function across all ovarian cell types. Strikingly, infectious ERV lineages display distinct expression patterns in somatic cells, from where they infect the germline, whereas derived retroelement lineages that have lost infectivity are expressed exclusively in the germline. Co-evolutionary changes in the piRNA pathway, which integrates transposon promoter and sequence information into differentially expressed piRNA clusters, highlight the functional significance of the diverse ERV expression niches. By investigating a unique ERV lineage, *rover*, we reconstruct the molecular events that transformed an infectious ERV into a retroelement. Overall, our study uncovers fundamental mechanisms that drive the co-evolution of ERVs and their hosts, with important implications for understanding the functional diversification of LTR sequences.**

**Keywords** Endogenous Retroviruses; Transposon Evolution; PIWI/piRNA Pathway; piRNA Cluster Evolution; *Drosophila* Germline Biology
**Subject Categories** Evolution & Ecology; Microbiology, Virology & Host Pathogen Interaction; RNA Biology

See also: R Halbach and RP van Rij

## Introduction

In the ancient genetic conflict between transposable elements (TEs) and their hosts, TEs seek to multiply within the host genome, while hosts evolve strategies to suppress TE activity. With such defense mechanisms in place, host organisms tolerate increasing amounts of TEs in their genomes, with profound evolutionary implications (Fedoroff, 2012; Kidwell, 2002; Kidwell and Lisch, 2001). However, while the influence of TEs on host genome evolution is increasingly well understood (Bohne et al, 2008; Bourque, 2009; Bowen and Jordan, 2002), the forces driving TE diversification remain less clear.

To investigate TE evolution, we focused on the *Metaviridae*, also known as *gypsy/Ty3* elements, an ancient and diverse group of long terminal repeat (LTR) retrotransposons (Llorens et al, 2020; Stefanov et al, 2012). *Metaviridae* typically harbor two open-reading frames, *gag* and *pol*, which encode the capsid proteins and enzymes for replication and genome integration, respectively. In animals, *Metaviridae* replicate within germline cells, but are silenced by the PIWI interacting RNA (piRNA) pathway (Senti and Brennecke, 2010; Siomi et al, 2011).

Notably, one specific branch of the *Metaviridae*, the Errantiviruses or insect endogenous retroviruses (iERVs), stands out due to a key biological trait: infectivity (Stefanov et al, 2012; Terzian et al, 2001). This is rooted in the presence of a third open reading frame encoding F-type Envelope proteins related to those of DNA baculoviruses (Malik et al, 2000; Rohrmann and Karplus, 2001). These F-type glycoproteins enable the fusion of enveloped viral particles with host cellular membranes. Consequently, the acquisition of *env-F* has transformed an intracellular replicating LTR retroelement into an infectious retrovirus. The evolutionary and biological consequences of this transformation remain largely unexplored.

Structurally, iERVs resemble members of the phylogenetically distinct alpha clade of *Retroviridae*, commonly found in vertebrates. Yet, unlike *Retroviridae*, iERVs do not spread as horizontally transmitted exogenous viruses. Instead, studies on two iERVs, *gypsy* and *ZAM*, have shown that they are expressed in follicle cells of *Drosophila* ovaries when the piRNA pathway is partially compromised. In these cases, the primary genomic piRNA source locus, *flamenco*, lacks sequences targeting specific iERVs, allowing

[1]Institute of Molecular Biotechnology of the Austrian Academy of Sciences (IMBA), Vienna BioCenter (VBC), Dr. Bohr-Gasse 3, 1030 Vienna, Austria. [2]Vienna BioCenter PhD Program, Doctoral School of the University of Vienna and Medical University of Vienna, Vienna, Austria. [3]University of St Andrews, Centre for Biological Diversity, St Andrews, Scotland, UK. [4]Institut für Populationsgenetik, Vetmeduni Vienna, Veterinärplatz 1, 1210 Vienna, Austria. ✉E-mail: senti@imba.oeaw.ac.at; julius.brennecke@imba.oeaw.ac.at

their expression (Brasset et al, 2006; Leblanc et al, 2000; Pelisson et al, 1994; Song et al, 1994). Once expressed, these iERVs infect the neighboring oocyte as virus-like particles and integrate new copies of themselves into the germline genome.

In this study, we use *Drosophila melanogaster* as a model system to investigate how an ancestral infectious retrotransposon diversified into the expansive iERV clade, and how the character of infectivity influenced this process. Through a comprehensive analysis of iERV genomes, their phylogenetic relationships, and their unique functional traits in vivo, we define key principles that governed the evolution of iERVs. Our findings shed light on the adaptive features underlying the niche-specific expression and replication strategies of iERVs and provide important insights into the intricate co-evolution between endogenous retroviruses and their hosts.

## Results

### The monophyletic iERV clade comprises both retroviruses and derived retroelements

The *Drosophila melanogaster* genome contains insertions of several dozen *Metaviridae* lineages, each defined by a distinct consensus sequence (Kaminker et al, 2002; Kapitonov and Jurka, 2003). We systematically curated and annotated all major open reading frames (ORFs) for these lineages and estimated a phylogenetic tree based on a full-length *pol* protein sequence alignment, using LTR-retroelements from the structurally related *Belpaoviridae*, which are distinct in their origin, as an outgroup (Krupovic et al, 2018). This analysis divided the *Metaviridae* into five recognized clades; *gypsy/mdg3, gypsy/osvaldo, gypsy/mdg1, gypsy/chimpo*, and *gypsy/gypsy* (Fig. 1A) (Kapitonov and Jurka, 2003). Seventeen *Metaviridae* lineages contain a third ORF, a predicted *envelope* gene (*env*), downstream of *gag* and *pol* (red dots in Fig. 1A,B). All of these lineages belong to the *gypsy/gypsy* elements, forming the most diverse *Metaviridae* clade also known as *Errantiviruses* or insect endogenous retroviruses (iERVs) (Bargues and Lerat, 2017; Stefanov et al, 2012; Terzian et al, 2001).

To comprehensively characterize the *envelope* ORFs of Metaviridae, we considered that Env is typically translated from a spliced sub-genomic transcript (Fig. 1B) (Leblanc et al, 2000; Marsano et al, 2000; Pelisson et al, 1994; Shigenobu et al, 2006; Tcheressiz et al, 2002). We experimentally identified the splice junctions for structurally intact lineages and predicted those for structurally defective lineages (see below). The main *env* exon is spliced to different upstream exons depending on the iERV subclade: from a short peptide-encoding exon upstream of *gag* in the *ZAM* and *Beagle* subclades, from an AU dinucleotide in the *springer* subclade, or from within *gag* in the *idefix* subclade (Appendix Figs. S1 and S2). Sequence alignment of the seventeen full-length iERV Envelope proteins confirmed their unambiguous similarity to F-type Envelope glycoproteins of baculoviruses (Appendix Fig. S3). The most conserved motifs include those with known functions in baculovirus Envelope proteins, such as the furin cleavage site, the fusion peptide, the transmembrane domain, and conserved cysteine residues that likely form stabilizing di-sulfide bonds (Fig. 1C; Appendix Fig. S3) (Malik et al, 2000; Rohrmann and Karplus, 2001). We refer to this third ORF in iERVs as *env-F*.

Using the complete set of curated iERV consensus sequences, we conducted systematic phylogenetic analyses. Trees estimated independently using two maximum likelihood methods from sequence alignments of full-length Pol or the more rapidly diverging Gag core domain showed congruent phylogenetic architectures (Fig. 1D; Appendix Fig. S4A,B). However, eleven iERV lineages in different subclades lack a functional *env-F* ORF, displaying different degrees of *env-F* degradation (Fig. 1D, Appendix Fig. 5). The lineage with the least degree of degradation, *rover*, comprises variants with and without intact *env-F* (see below). We therefore estimated a phylogenetic tree from a sequence alignment of all seventeen full-length Env-F proteins. This tree mirrored the Pol and Gag trees, arguing against multiple independent events of *env-F* gain or *env-F* exchange by recombination between iERV lineages (Appendix Fig. S4C). Notably, we found that lineages lacking *env-F* often contain sequences of variable length between *pol* and the 3' LTR (Fig. 1E). Most of these sequences retain mutated but recognizable *env-F* fragments (Appendix Figs. S5 and S6), strongly suggesting that iERV lineages lacking *env-F* evolved multiple times from infectious retroviruses through the degradation of their *env-F* genes.

We then explored the origin of the character state of functional *env-F* in the phylogenetic tree and its influence on iERV diversification. Using the binary character state speciation and extinction model (BiSSE) (FitzJohn et al, 2009), we estimated a 99.1% probability that the ancestral state of the iERV clade included a functional *env-F* (Appendix Fig. S7A). Bayesian analysis indicated that retroviruses (with *env-F*) diversified at a higher rate than related retroelements lacking *env-F* (Appendix Fig. S7B), though this difference did not reach statistical significance when based on the limited number of iERV lineages in *Drosophila melanogaster* (Davis et al, 2013).

We finally examined the *env-F* status alongside the integrity of the *gag* and *pol* ORFs within all iERV insertions found in the *D. melanogaster* reference genome (dm6). This classified iERV lineages as either *env-F*-encoding retroviruses (12 active & 4 inactive lineages) or *env-F*-deficient retroelements (7 active & 4 inactive lineages) (Fig. 1D). All inactive retrovirus and retroelement insertions are located in the recombination-poor pericentromeric heterochromatin (Talbert and Henikoff, 2010) and almost all show higher pairwise LTR divergence compared to their structurally intact relatives (Appendix Fig. S7C), indicating that they are sequence fossils of once active iERV lineages. In summary, our results provide strong evidence that iERVs form a monophyletic clade of LTR retrotransposons, derived from an ancestral retrovirus containing *env-F*, and encompass a wide array of retroviral and derived, non-infectious retroelement lineages.

### Infectious and non-infectious iERVs exhibit distinct expression and replication strategies

Building on the evolutionary relationships among iERV lineages, we investigated the link between infectivity (presence/absence of functional *env-F*) and expression patterns. Our focus was on the ovary of adult flies, where germline and somatic cells maintain direct contact throughout oogenesis (Duhart et al, 2017; Kirilly and Xie, 2007). In wildtype flies, distinct piRNA pathways silence TEs in the ovarian soma and germline, respectively (Malone et al, 2009; Senti and Brennecke, 2010). To uncover the spatio-temporal

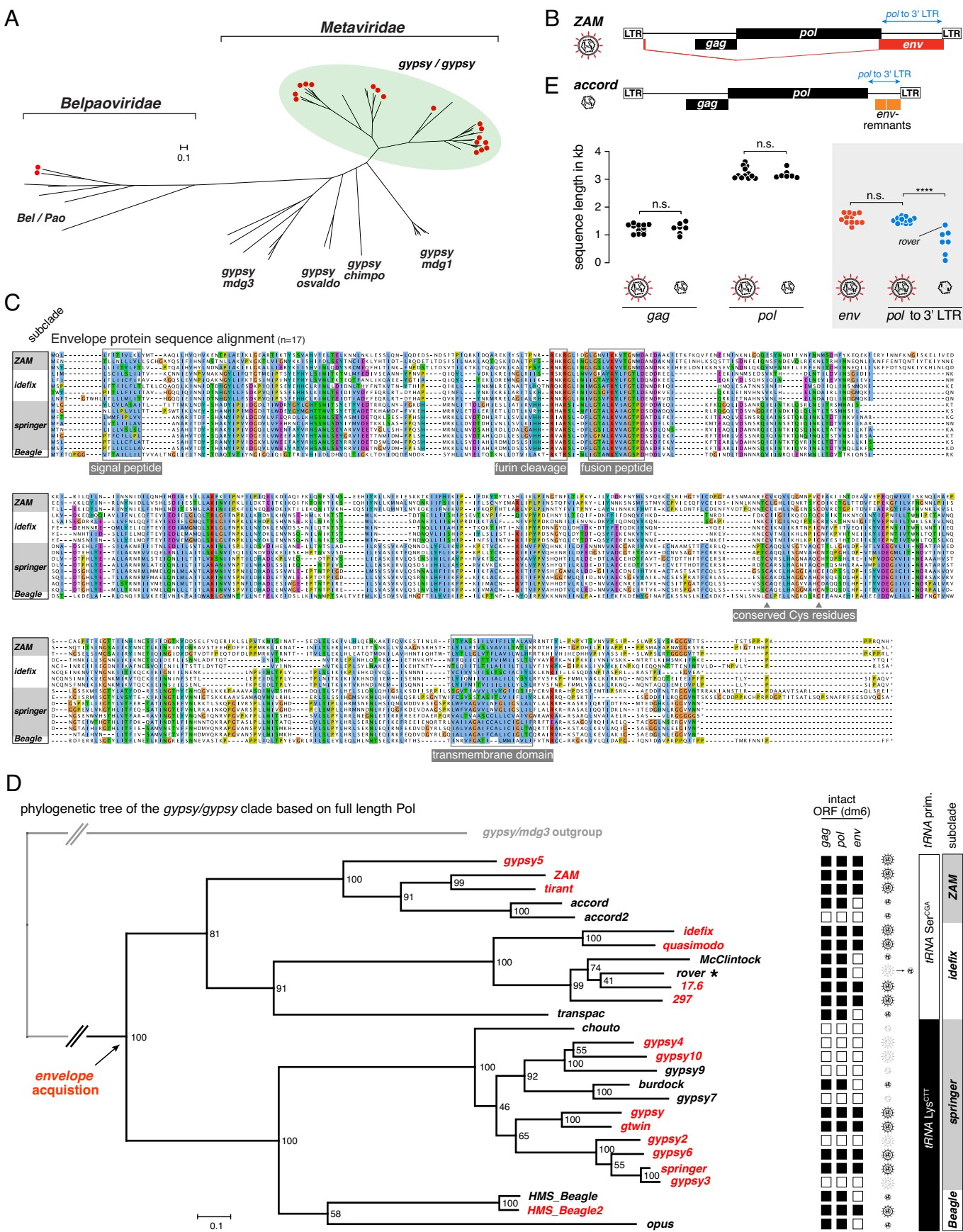

**Figure 1. Phylogeny and evolution of iERVs in *Drosophila melanogaster*.**

(A) Phylogenetic tree (scale indicates amino acid substitutions per site), based on full-length Pol, of all *Belpaoviridae* and *Metaviridae* LTR-retrotransposon consensus sequences from *Drosophila melanogaster*. LTR element subclades are indicated and consensus sequences with a full-length *env-F* gene are marked with a red dot. (B) Cartoon depicting the structure of an infectious iERV (*ZAM*) indicating the spliced *env-F* ORF (red). (C) Protein sequence alignment of all spliced Envelope-F proteins from *Drosophila melanogaster* iERVs with N-terminal signal peptide, furin cleavage site, fusion peptide, conserved Cysteine residues presumably involved in di-sulfide bond formation, transmembrane domain, and C-terminal cytoplasmic tail highlighted. (D) Phylogenetic tree (scale indicates amino acid substitutions per site), based on full-length Pol, of all iERVs from *Drosophila melanogaster* (outgroup: *gypsy/mdg3* clade; numbers indicate bootstrap values). Lineages with full-length *env-F* in their consensus sequence are labeled in red, retroelement revertants in black. To the right, the structural integrity of each lineage based on intact *gag, pol, env-F* open reading frames in at least one insertion in *dm6* is shown. Retroviral and retroelement symbols indicate active (black) or inactive (gray) lineages (asterisk: transition element *rover*). tRNA priming and subclades are indicated to the right. (E) Cartoon depicting the structure of a retroelement revertant (*accord*) indicating *env* remnants (orange). The jitter plot below shows the length (in kb) of *gag, pol, env-F* open reading frames as well as the *pol* to 3′ LTR distance for all active iERVs (statistical significance based on unpaired, non-parametric two-tailed Mann–Whitney test, **** indicates a significant *P* value of <0.0001, n.s. indicates statistically non-significant differences).

expression patterns of iERVs, we utilized tissue-specific, transgenic RNA interference (Dietzl et al, 2007; Handler et al, 2013; Ni et al, 2011; Senti et al, 2015). Using *tj*-Gal4 and long double-stranded RNA transgenes (GD-lines), we depleted key piRNA pathway factors, Vreteno or Zucchini, in all ovarian somatic cells (soma-knockdown). Using MTD-Gal4 and short hairpin transgenes (sh-lines), we depleted the essential factors Spindle-E, Aubergine or Aubergine/Ago3 in all ovarian germline cells (germline-knock-down) (Appendix Fig. S8). PolyA⁺ RNA-seq experiments revealed that transcript levels of active retroviruses were de-repressed in soma-knockdown ovaries but not in germline-knockdown ovaries, whereas active retroelements were specifically de-repressed in germline-knockdown ovaries (Fig. 2A,B; Appendix Fig. S9A,B). The retrovirus *tirant*, which naturally invaded *D. melanogaster* populations in the 20th century, could not be conclusively assessed due to its absence in most experimental strains. These findings suggested that active retroviruses and retrovirus-derived retroelements evolved differential expression patterns in ovarian tissues, with retroviruses being active in somatic cells and retroelements in germline cells.

Despite their tissue-specific transcription, iERVs must integrate new copies of themselves into the germline genome. To investigate their replication strategies, we employed single-molecule fluorescent RNA in situ hybridization (smFISH) (Raj et al, 2008; Senti et al, 2015), focusing on two representative iERVs: the retrovirus *ZAM* and the retroelement *McClintock* (Appendix Fig. S9B). In control ovaries and in ovaries lacking piRNA control in the germline (germline knockdown ovaries), *ZAM* transcripts were undetectable. In contrast, *ZAM* was expressed at high levels in ovaries lacking somatic piRNA control (soma knockdown ovaries), peaking in stage 9/10 egg chambers (Fig. 2C; Appendix Fig. S10A,B) (Leblanc et al, 2000). Here, *ZAM* transcripts were enriched at the apical membrane of posterior follicle cells, but also at the adjacent oocyte membrane, and within the oocyte, despite the lack of germline transcription. Within the oocyte, *ZAM* transcripts accumulated in crescent-shaped structures at the periphery of yolk granules labeled by GFP-tagged Yolk Protein 1 (Fig. 2D; Appendix Fig. S10C) (Hara and Yamamoto, 2021; Raikhel and Dhadialla, 1992), supporting the observation that *ZAM* hijacks the vitellogenesis pathway for transfer into the oocyte (Brasset et al, 2006). To quantify germline transmission, we sequenced poly-adenylated RNA from dechorionated 0–1 h old embryos, which contain only maternally supplied germline RNAs. In embryos laid by mothers with defective somatic piRNA control, *ZAM* transcript levels were ~30-fold higher than in controls (Appendix Fig. S9). *ZAM* RNA localized to disintegrating yolk granules but also to the posterior pole (Fig. 2E; Appendix Fig. S10D).

We observed a similar soma-to-oocyte transfer for the *springer* and *gypsy* retroviruses (Appendix Fig. S11A–D).

In contrast to *ZAM*, the non-infectious retroelement *McClintock* was exclusively transcribed in germline knockdown ovaries. *McClintock* transcripts and its capsid protein, Gag, were detected in nurse cells and enriched in the transcriptionally inactive oocyte, suggesting transport of *McClintock* capsids from nurse cells into the oocyte through ring canals (Fig. 2F; Appendix Fig. S10E–G) (Senti et al, 2015; Wang et al, 2018). *McClintock* transcript levels increased ~150-fold in 0–1 h old embryos laid by mothers with germline knockdown compared to controls (Appendix Fig. S9C). Similar to *ZAM*, *McClintock* transcripts and those of other retroelements (e.g., *burdock* and *rover*) accumulated at the posterior pole of embryos (Fig. 2G; Appendix Figs. S10H and S11E–G). Thus, despite originating from different tissues, both *McClintock* and *ZAM* target their genetic material to the posterior pole plasm, where the primordial germ cells of the embryo will develop.

## Niche specialization among infectious iERVs in the ovarian soma

The increase in transcript levels of iERVs in soma-knockdown ovaries varied widely, with fold changes ranging from 3-fold (*17.6*) to ~600-fold (*ZAM*) (Appendix Fig. S9A). To determine whether this variation resulted from temporally or spatially restricted iERV expression domains, we conducted RNA smFISH for all active retroviruses throughout adult oogenesis. Our analysis revealed that, while repressed in control ovaries, each retroviral lineage exhibited a distinct expression pattern in soma-knockdown ovaries. Collectively, retroviruses occupied nearly every cell type in the ovarian soma (Fig. 3A summarizes the expression data shown in Appendix Fig. S12). Retroviruses with strong de-repression at the RNA-seq level (*ZAM, gypsy, springer, gypsy6, HMS-Beagle 2*) were expressed in broad domains, while those with moderate de-repression (*idefix, 17.6, quasimodo, 297, gypsy5*) exhibited more restricted expression niches (Appendix Fig. S12). Moreover, viruses from the *idefix* subclade were predominantly expressed during early oogenesis, whereas members of the *ZAM, springer,* and *Beagle* subclades were active mainly at late stages (Fig. 3A; Appendix Fig. S12). This suggested that different iERV subclades target the germline genome at distinct oogenesis stages.

Notably, several related retroviral lineages sharing a common ancestor exhibited spatially adjacent but distinct expression patterns. For instance, the *idefix/quasimodo* pair was expressed in non-overlapping cell populations of the germarium, with *idefix* in terminal filament and cap cells, and *quasimodo* in escort cells

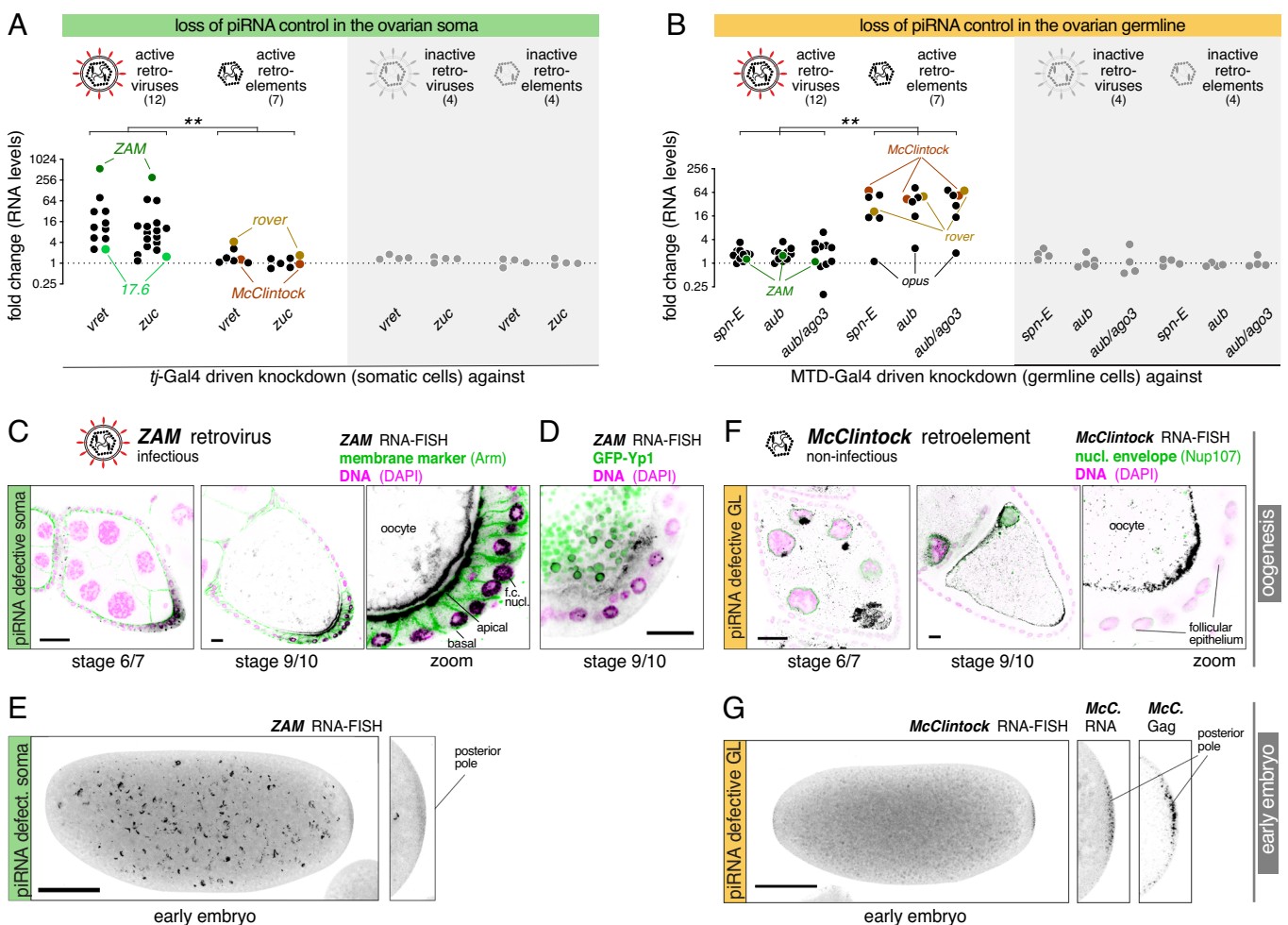

**Figure 2. Distinct replication strategies of infectious and non-infectious iERVs in *Drosophila*.**

(**A, B**) Shown are fold changes in ovarian steady state levels of iERV transcripts, grouped into active and inactive retroviruses and retroelements, upon loss of somatic piRNA pathway versus control (**A**) (*tj*-Gal4 driven long dsRNA hairpins against *vreteno* or *zucchini*) or loss of germline piRNA pathway versus control (**B**) (*MTD*-Gal4 driven shRNAs against *spn-E*, *aub*, or *aub+ago3*). Two-tailed Mann–Whitney statistical test; ** indicate P values of <0.01 between active retrovirus and active retro-element expression in the same genotypes. (**C**) RNA-smFISH based expression analysis for the retrovirus *ZAM* in egg chambers of indicated stage from ovaries with defective somatic piRNA pathway (*tj*-Gal4 driven dsRNA hairpin against *vreteno*; scale bars: 20 µm; cell outlines stained with anti-Armadillo and DNA with DAPI). The magnified panel shows the posterior pole of the growing stage 9/10 oocyte with adjacent follicle cells (f.c.). (**D**) As in (**C**) but with yolk granules labeled with YP1-GFP in green. (Scale bar: 20 µm). (**E**) RNA-smFISH based expression analysis for *ZAM* in pre-blastoderm embryos with less than 32 nuclei laid by females lacking somatic piRNA control (genotypes: *tj*-Gal4 > *vreteno^GD*). (**F**) RNA-smFISH based expression analysis for the retroelement *McClintock* in egg chambers of indicated stage from ovaries with defective germline piRNA pathway (*MTD*-Gal4 driven shRNA against *aub+ago3*) and GFP-nup107 in green. (**G**) As in (**F**) but RNA-smFISH based expression analysis for *McClintock* in embryos laid by females lacking germline piRNA control (genotype: *MTD*-Gal4 > shRNA^aub). Images in (**E**) and (**G**) show maximum intensity Z projections (scale bars: 100 µm). Magnified panels show accumulation of FISH or anti-Gag signal at the posterior pole.

(Fig. 3B). Within the same subclade, *17.6* and *297* showed expression in cap cells, escort cells, and early follicle cells (*17.6*) or in escort cells, early follicle cells, and polar cells (*297*) (Appendix Fig. S13A,B). Divergent expression patterns were also observed for related retroviruses active during later oogenesis stages: *ZAM* was expressed in posterior follicle cells and border cells, while *gypsy5* was mainly expressed in border cells and polar cells (Fig. 3C). Similarly, *gypsy* was expressed in all main body follicle cells of late-stage egg chambers, while the related *gtwin* virus was largely restricted to centripetal cells (Appendix Fig. S13C,D). Lastly, *springer* and *gypsy6* showed overlapping patterns in main body follicle cells but initiated expression at different stages (stage 6/7

versus stage 9/10, respectively) (Appendix Fig. S13E,F). These observations suggest that infectious iERV lineages collectively occupy the full spectrum of ovarian somatic cell types, and that competitive processes (see discussion) may drive the divergence of expression niches among related retroviral lineages (Fig. 3A).

## Secondary loss of infectivity strictly correlates with germline expression of iERVs

The distinct expression niches of infectious iERVs in the ovarian soma raise the question of how loss of infectivity has influenced the evolution of non-infectious retroelements. Functional *env-F* has

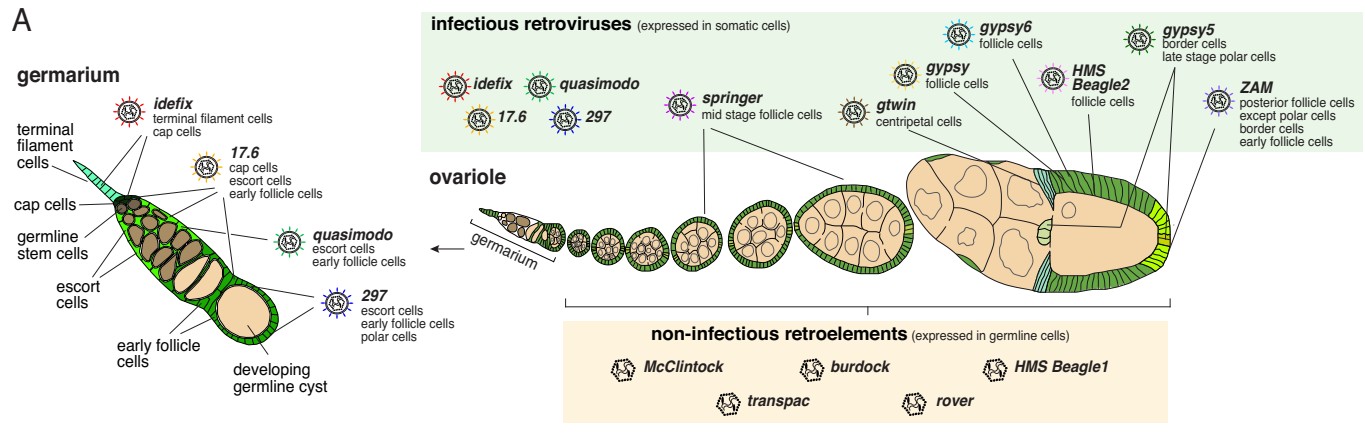

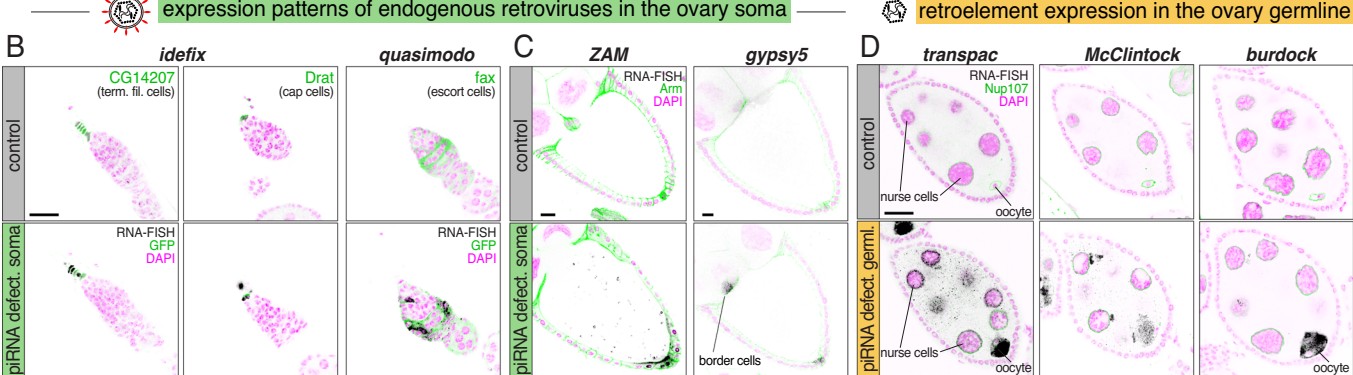

**Figure 3. Pervasive niche expression among infective and non-infective iERVs.**

(A) Cartoon summarizing the major expression domains of infective and non-infective iERV lineages in the germarium (left) and entire ovariole (right). Note that their expression is only visible in piRNA-deficient ovaries. Germline cells are shown in beige, somatic cells in green. (B, C) RNA-smFISH based expression analysis of closely related retroviral pairs in control ovaries or ovaries lacking somatic piRNA pathway control (*tj*-Gal4 driven dsRNA hairpins against *arrestin2* or *vreteno*). (B) Shows *idefix* and *quasimodo* (black) in the germarium with indicated cell types identified by specific GFP-trap lines (green). (C) Shows *ZAM* and *gypsy5* expression (black) in stage 9/10 egg chambers with cell outlines marked by anti-Armadillo (green). *gypsy5* images in (C) show maximum intensity projections of five Z-sections. Scale bars: 20 μm. (D) RNA-smFISH based expression analysis of three retroelement revertants (black) in stage 6/7 egg chambers of control ovaries or ovaries lacking germline piRNA control (*MTD*-Gal4 driven shRNAs against *white* or *aub+ago3*; GFP-Nup107 (green) labels nuclear envelopes). Scale bars: 20 μm.

been lost multiple times during iERV diversification, resulting in ten clear retroelement lineages, six of which remain active (*accord, McClintock, transpac, burdock, HMS-Beagle, opus*) (Fig. 1D). These retroelement lineages displayed strong expression in germline knockdown ovaries but were not expressed in soma knockdown ovaries (Fig. 2A,B). RNA smFISH experiments confirmed the RNA-seq results, indicating that iERV retroelements are specifically expressed in germline cells, leading to the accumulation of transcripts in the maturing oocyte (Fig. 3D; Appendix Fig. S14A–C). Retroelement expression commenced in differentiating cystoblasts and persisted through late-stage oogenesis (Appendix Fig. S15). One exception was *opus*, which was not expressed in either the soma or germline piRNA pathway knockdown ovaries. Notably, smFISH experiments in piRNA pathway-deficient testes revealed that *opus* is instead expressed in the differentiating male germline (Appendix Fig. S14D). Together, our results indicate that infectious retroviruses occupy different niches within the ovarian soma, while derived, non-infectious retroelements have lost somatic expression and gained germline-specific expression.

## iERV-intrinsic *cis*-regulatory sequences define somatic and germline expression niches

The distinct expression patterns observed among iERVs may arise from *cis*-regulatory sequences intrinsic to the retrotransposons themselves or from dominant host genome enhancers located nearby individual iERV insertions. Sequence comparisons indicated that the LTR and 5′ UTR regions—known to harbor *cis*-regulatory elements in LTR retroviruses and retroelements (Johnson, 2019)—are the most divergent sequences among iERVs (Fig. 4A; Appendix Fig. S16A). Only short sequence motifs at the LTR borders (likely integrase recognition motifs) (Appendix Fig. S16B) and at the beginning of the 5′ UTR (part of the primer binding site for specific tRNAs) (Terzian et al, 2001) were conserved. To test whether the cell type-specific transcription of iERVs is driven by their non-coding sequences, we examined lacZ reporter fly lines for the retroviruses *idefix* and *ZAM* (Desset et al, 2003; Tcheressiz et al, 2002), and generated new reporter lines for the retroviruses *quasimodo* and *gypsy5*, as well as for the retroelements *burdock*,

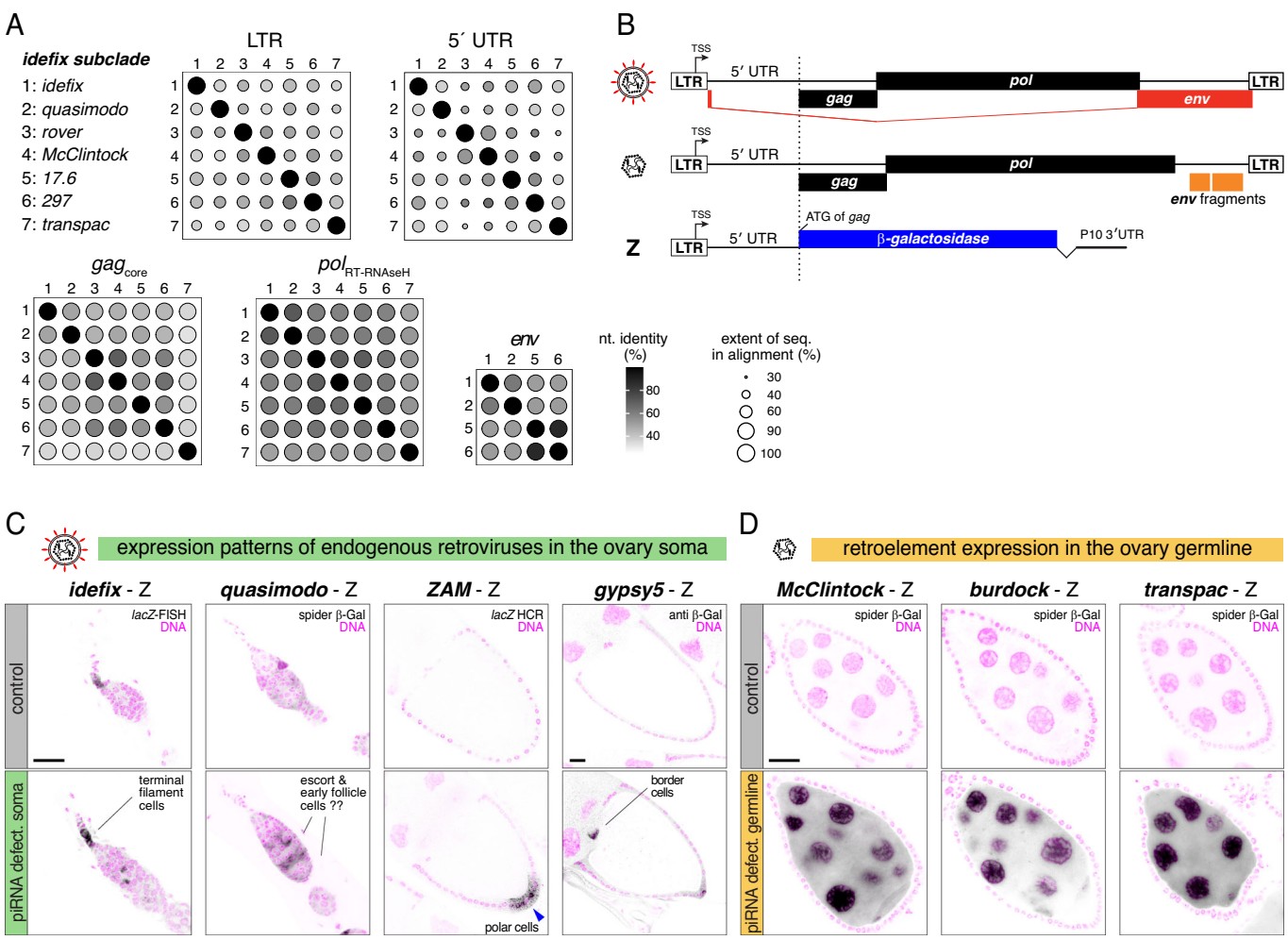

**Figure 4. iERV-intrinsic *cis*-regulatory sequences define somatic and germline expression niches.**

(A) Shown are pairwise identities between the DNA sequences (*idefix* subclade) for LTR, 5'UTR, and ORFs for Gag_core, Pol_RT-RNAseH, and Env-F (circle size indicates the percentage of nucleotides in aligned positions between each two sequences, gray scale indicates pairwise nucleotide identities between iERVs in the alignment). (B) Cartoon illustrating the logic of lacZ-reporter transgenes for retroviruses and retroelements. (C, D) Analysis of retrovirus (C) or retroelement (D) lacZ reporter expression in control ovaries (top row) and in ovaries lacking somatic or germline piRNA pathway control (genotypes: *tj*-Gal4 driven dsRNA hairpins against *vreteno* or *arrestin2* (C); MTD-Gal4 driven shRNAs against *aub* or *white* for *McClintock* and *burdock* and against *aub*+*ago3* for *transpac* (D)); germarium stages are shown for *idefix* and *quasimodo*, stage 9/10 egg chambers for *gypsy5* and *ZAM*, and stage 6/7 egg chambers for retroelements; scale bars: 20 µm.

*McClintock*, and *transpac* (Fig. 4B). In wildtype flies, all reporters showed no or only weak expression, consistent with PIWI/piRNA-mediated repression. In soma knockdown ovaries, all four retroviral reporters—but none of the retroelement reporters—were expressed, with their patterns closely matching those of their endogenous counterparts (Figs. 3B,C and 4C). Conversely, in germline knock-down ovaries, only the retroelement reporters were active, again mirroring the expression of their respective endogenous counterparts (Figs. 3D and 4D). These findings demonstrate that two iERV traits—infectivity status (*env-F* coding potential) and intrinsic *cis*-regulatory sequences—co-evolve. While infectious iERV lineages have adapted their LTR and 5' UTR sequences for expression in specific somatic cell types, all non-infectious lineages have evolved *cis*-regulatory elements favoring germline-specific transcription. To determine whether these distinct expression patterns hold functional importance for the different iERV lineages, we next sought to

identify signatures of co-evolution within the PIWI/piRNA pathway, which is central to silencing iERV expression in gonads.

## piRNA clusters co-evolve with iERVs through the co-option of retroviral sequences

Two distinct Piwi/piRNA pathways operate in the soma and germline of the *Drosophila* ovary, each relying on different genomic piRNA source loci (Malone et al, 2009; Senti and Brennecke, 2010). In somatic cells, piRNA clusters are canonical RNA polymerase II transcription units with defined promoters (Goriaux et al, 2014; Mohn et al, 2014). The silencing spectrum of the somatic piRNA pathway depends on TE insertions within a piRNA cluster, oriented antisense to the cluster's unidirectional transcription. The primary piRNA cluster in the soma, *flamenco*, is a several hundred kilobase locus that is transcribed from a single promoter. iERV sequences

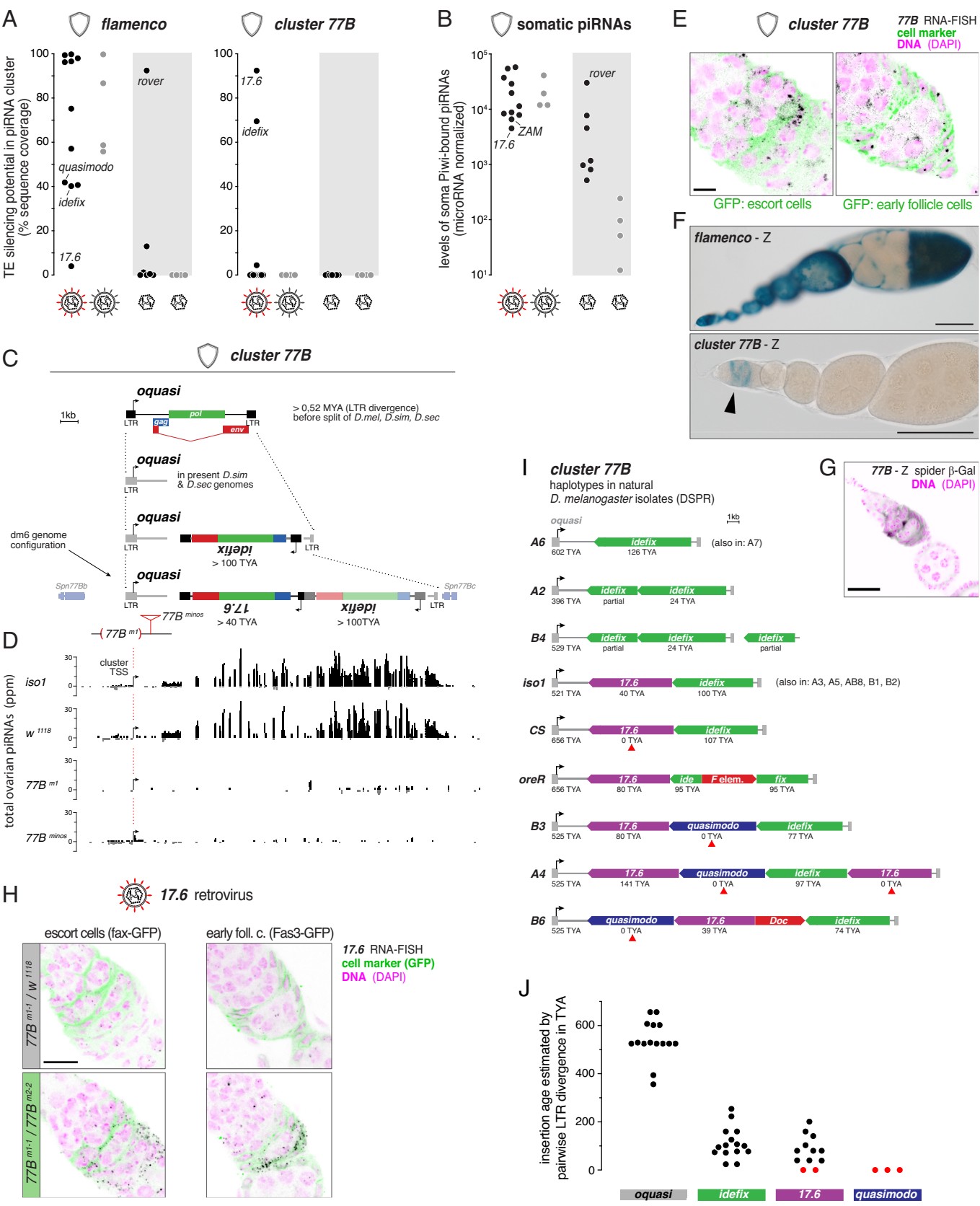

◄ **Figure 5. Co-evolution of the somatic piRNA pathway with infectious iERVs.**

(A) Shown is the silencing potential within indicated somatic piRNA clusters against active and inactive iERVs (retroviruses and retroelements) as fraction of TE sequence found as 25mers (no mismatches) in antisense orientation in the cluster. (B) Shown are microRNA-normalized levels of somatic, Piwi-bound piRNAs antisense to active or inactive iERV retroviruses and retroelements (no mismatches). (C) Proposed evolutionary trajectory of the *D. melanogaster* specific *77B* piRNA cluster that originated from an old *quasimodo* retrovirus (*oquasi*) insertion between the *Spn77Bb* and *Spn77Bc* genes and captured *idefix* and *17.6* retroviral insertions in antisense orientation (indicated insertion ages based on pairwise LTR divergence estimates). (D) Shown are microRNA-normalized levels of ovarian piRNAs from indicated wildtype strains (*iso1*, *w^1118^*) and cluster *77B* mutants (*77B^m1^* and *77B^minos^*). The transcription start site (TSS) of the cluster within the *oquasi* LTR is indicated (image in scale and aligned to panel (C)). (E) RNA-smFISH based expression analysis of *cluster 77B* (black) in the germarium. GFP-trap lines indicate escort cells and early follicle cells, respectively (scale bar: 10 µm). (F) Shown are X-Gal stainings of ovaries harboring lacZ-reporter transgenes for *flamenco* or cluster *77B* (*oquasi* LTR-5′UTR) (arrowhead indicates specific signal in the germarium; scale bars: 100 µm). (G) Detailed expression analysis of the cluster *77B* (*oquasi* LTR) lacZ-reporter expression (black) in the germarium with DNA (Hoechst) in magenta. (H) RNA-smFISH based expression analysis of the *17.6* retrovirus (black) in germaria of *77B^m1^/w^1118^* control (top) ovaries or of cluster *77B* mutant ovaries (cluster *77B^m1^/^m2^* trans-heterozygotes; bottom; scale bar: 10 µm). GFP-trap lines (green) mark indicated cell types. (I) Sequence content of the cluster *77B* in indicated DSPR strains. Below each annotated LTR element the estimated age of the insertion, based on pairwise LTR divergence, is indicated (TYA: thousand years ago). Red arrowheads mark recent insertions (modern *quasimodo* and *17.6*) with no LTR divergence. (J) Jitter plot showing the pairwise LTR divergence estimated insertion age of the *oquasi*, *17.6*, *idefix*, and *quasimodo* insertions within the cluster *77B* of all analyzed DSPR strains (red dots mark recent insertions with no LTR divergence).

are notably enriched in *flamenco* (Goriaux et al, 2014; Malone et al, 2009; van Lopik et al, 2023), and flies carrying specific permissive *flamenco* alleles exhibit de-repression of the retroviruses *gypsy*, *ZAM*, and *idefix* (Brennecke et al, 2007; Desset et al, 1999; Sarot et al, 2004). These findings suggest that a single locus, *flamenco*, might be responsible for silencing the entire spectrum of infectious iERVs across the ovarian soma.

To test the "single cluster" hypothesis, we first examined *flamenco* expression in the ovary using smFISH. We found that *flamenco* is transcribed in all somatic cell types throughout oogenesis, except in terminal filament cells (Appendix Fig. S17A). Next, we analyzed the sequence composition of *flamenco* in relation to the iERV phylogenetic tree, determining the proportion of each iERV sequence represented as antisense 25-mers (indicative of silencing potential). This analysis revealed a striking bias in *flamenco*'s iERV content (Zanni et al, 2013): while infectious retroviruses are strongly represented regardless of their activity status, non-infectious iERV lineages derived from retroviruses were notably absent (Fig. 5A). The only exception was *rover*, a transition lineage examined further below.

Interestingly, *flamenco* exhibits only moderate silencing potential against the infectious iERVs *idefix* and *quasimodo*, and none against *17.6* (Fig. 5A). Despite this, wildtype ovaries produce abundant antisense piRNAs targeting *17.6* in the ovarian soma (Fig. 5B), and the *17.6* retrovirus is repressed by the piRNA pathway (Appendix Fig. S13A). This discrepancy pointed to the existence of an additional piRNA cluster involved in *17.6* silencing. We identified a promising candidate on chromosome 3 L, cytological position 77B, which contains three iERV insertions from the *idefix* subclade (Fig. 5C) (Chen and Aravin, 2023). This locus, termed *cluster 77B*, harbors a partial copy of an old *quasimodo*-related element (termed *oquasi*) and one insertion each of *17.6* and *idefix*, both inserted in opposite orientation to *oquasi* (Fig. 5C). Small RNA-seq data revealed that *cluster 77B* is unidirectional and generates piRNAs associated with somatic Piwi but not with germline Piwi/Aub/Ago3 (Fig. 5D; Appendix Fig. S18B–F). *cluster 77B* is also active in cultured ovarian somatic cells (OSCs), and RNA-seq, PRO-seq, and piRNA-seq data all indicate that its transcription initiates from the initiator motif in the first *oquasi* LTR, producing a polyadenylated transcript ~17 kb in length (Appendix Fig. S17G).

Unlike the broad expression of *flamenco*, *cluster 77B* is specifically expressed in escort cells and early follicle cells of the germarium (Fig. 5E), mirroring the expression of the *quasimodo* retrovirus (Appendix Fig. S12). To validate this pattern, we generated reporter transgenes for *flamenco* and *cluster 77B*, containing the lacZ ORF downstream of their respective promoter regions. While the *flamenco* reporter was active in all somatic cells of the ovary, the *cluster 77B* reporter recapitulated the expression pattern of the endogenous piRNA cluster in the germarium (Fig. 5F,G). These results suggest that *D. melanogaster* has co-opted the LTR and 5′ UTR sequences of a fixed *oquasi* retrovirus to drive its own silencing program.

The expression domain of *cluster 77B* overlaps with that of the *17.6* retrovirus, for which *flamenco* lacks silencing capacity. To determine the functional significance of *cluster 77B*, we generated a mutant allele by deleting the *oquasi* LTR that serves as its transcription start site. Flies homozygous for this deletion, as well as flies carrying a Minos insertion within the *oquasi* 5′ region, showed loss of *cluster 77B* expression and corresponding piRNAs (Fig. 5C,D). In these mutants, *17.6* was de-repressed in escort cells and early follicle cells, demonstrating the critical role of *cluster 77B* in controlling this retrovirus (Fig. 5H; Appendix Fig. S17H).

The relatively small size and cell type-specific expression of *cluster 77B* provided a unique opportunity to study the evolution of a somatic piRNA cluster. Pairwise LTR divergence analysis estimated that the *oquasi* insertion, which is closely related to the *Nquasimodo* retrovirus from *Drosophila erecta* (Bargues and Lerat, 2017), originated over 500,000 years ago. The more recent *idefix* and *17.6* insertions occurred around 100,000 and 40,000 years ago, respectively. Consistent with these estimates, the syntenic genomic regions in *D. simulans* and *D. sechellia*, which share a common ancestor with *D. melanogaster* about 2.7 million years ago (Obbard et al, 2012), contain the *oquasi* fragment but lack *17.6* and *idefix* insertions (Fig. 5C). The absence of the *17.6* retrovirus in the genomes of *D. simulans* and *D. sechellia* suggests that *cluster 77B* evolved as a targeted defense mechanism in *D. melanogaster*. Further analysis of the genomes of natural *D. melanogaster* isolates that were collected around the globe, the DSPR founder strains (Chakraborty et al, 2019), revealed extensive structural variation of *cluster 77B*, indicating ongoing adaptive evolution. Notably, TE insertions in natural *cluster 77B* alleles are predominantly antisense insertions from the *idefix* subclade, precisely those lineages that are co-expressed with *cluster 77B* in the germarium (Fig. 5I,J).

In contrast to the relatively hard-wired piRNA clusters in the soma, germline cells generate TE-silencing piRNAs from transcripts

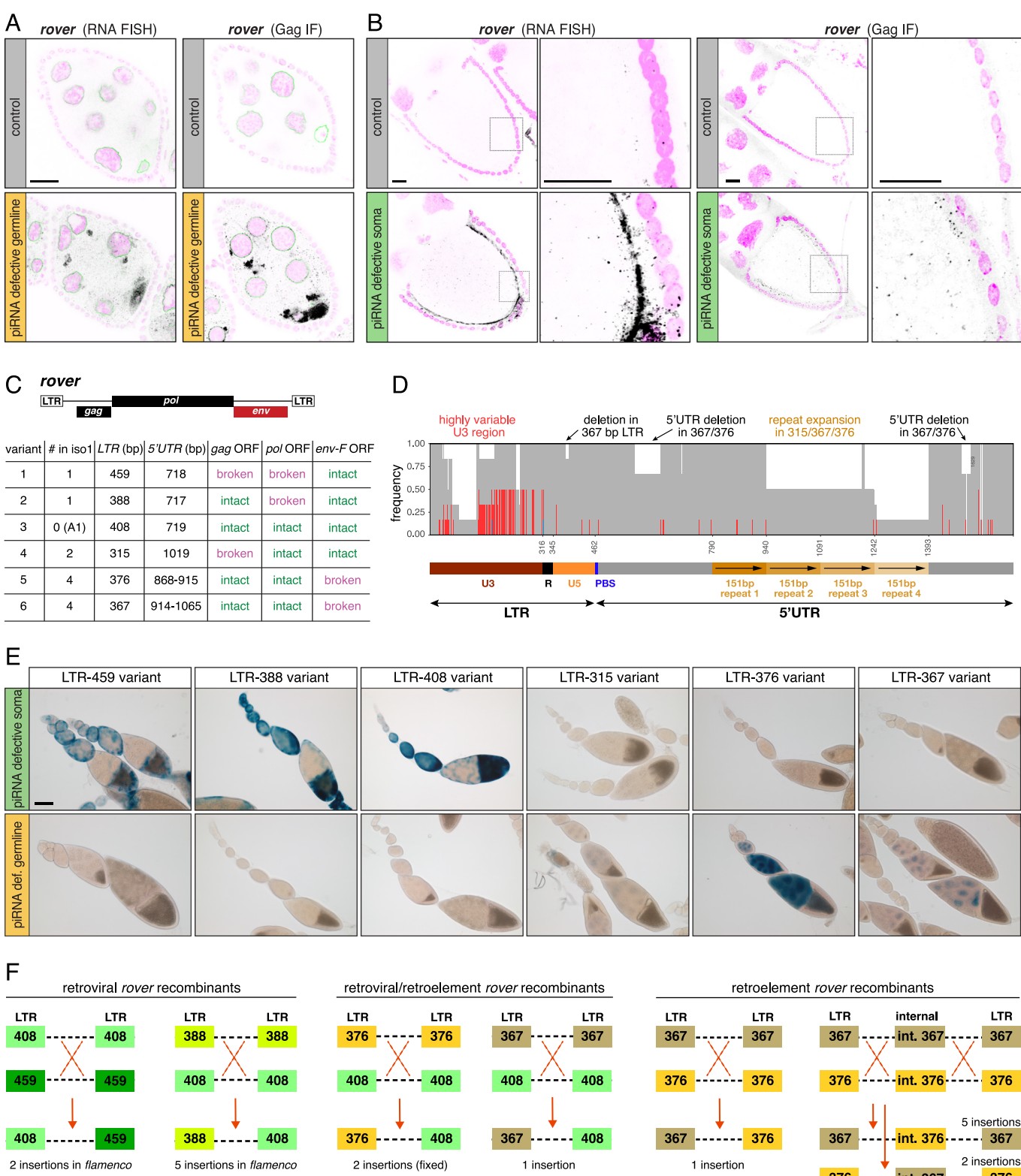

derived from hundreds of heterochromatic loci specified by the HP1-variant protein Rhino (Klattenhoff et al, 2009; Mohn et al, 2014; Zhang et al, 2014). In *rhino* mutant ovaries, levels of piRNAs targeting all the germline-expressed, non-infectious iERV lineages

were strongly reduced (Appendix Fig. S18A). While we found no corresponding iERV insertions or silencing potential (determined as 25mer sequences in cluster loci mapping to iERV sequences) in the most prominent piRNA clusters (Appendix Fig. S18B), several

**Figure 6. Diversity and retrovirus to retroelement evolution of the *rover* lineage.**

(A, B) *rover* RNA-smFISH and *rover* Gag staining in control and piRNA pathway-deficient ovaries (A germline pathway defective; B soma pathway defective; black: *rover* signal; magenta: DAPI; scale bars: 20 μm). (C) Summary of *rover* variants identified in the *iso-1* genome and a representative of the LTR-408 variant from the DSPR genome A1 (LTR and 5′ UTR length indicated in bp). (D) Frequency plot of SNPs across an alignment of the *cis*-regulatory sequences (LTRs and 5′ UTRs) of six representative *rover* variants. White (no label) represents no aligning sequence, gray represents the most frequent nucleotide frequency, red indicates the second most frequent nucleotide frequency, blue shows the third most frequent nucleotide frequency. The schematic below provides the annotations for the functional elements. (E) X-gal staining of ovaries from transgenic flies bearing *lacZ* reporter constructs encompassing *cis*-regulatory sequences of all six *rover* variants. Ovaries are defective either for the soma (top row) or germline piRNA pathway (bottom row). Scale bar: 100 μm. (F) Schematic of recombination events between different *rover* variants identified in DSPR genomes.

stand-alone iERV retroelement insertions, bound by Rhino and its co-factor Kipferl, functioned as piRNA source loci (Appendix Fig. S18C) (Baumgartner et al, 2022; Mohn et al, 2014; Olovnikov et al, 2013; Shpiz et al, 2014). These findings illustrate the adaptive flexibility of the germline piRNA pathway, where TE control relies on chromatin-based activation of loci containing TE insertions.

In conclusion, the extensive co-evolution between the piRNA pathway and both infectious retroviruses and non-infectious retroelements strongly suggests that the expression of iERV lineages in distinct somatic and germline niches plays a critical role in their replication cycles.

## The molecular basis underlying the transition from retrovirus to retroelement

Finally, we asked whether the sequence record of iERV insertions in the genomes of different *Drosophila melanogaster* isolates could help reconstruct the evolutionary transition from a retrovirus to a germline-expressed retroelement. In our earlier analysis (Fig. 1D; Appendix Fig. S9), the *rover* lineage stood out among the typically soma-restricted retroviruses and germline-expressed retroelements, as it exhibited characteristics of both: In germline piRNA-deficient ovaries, *rover* was strongly expressed in nurse cells, with transcripts and Gag proteins accumulating in the maturing oocyte (Fig. 6A). However, unlike other retroelements, *rover* was also expressed in ovaries with defective somatic piRNA pathway. The somatically expressed *rover* transcripts and Gag proteins resembled those of infectious retroviruses, being enriched at the apical follicle cell membrane and showing clear evidence of soma-to-oocyte transfer (Fig. 6B). Moreover, in contrast to typical infectious iERVs and non-infectious iERVs that are repressed in either the soma or the germline, respectively, both piRNA profiling data as well as piRNA cluster silencing potential indicated that *rover* is silenced in both soma and germline (Fig. 5A,B; Appendix Fig. S18A,C).

We hypothesized that *rover* either represents a lineage with both retroviral and retroelement characteristics or that distinct *rover* variants with exclusively retroviral or retroelement characteristics exist. To explore this, we analyzed *rover* sequences in the *D. melanogaster* reference genome (12 insertions) and in the long-read genome assemblies of the thirteen DSPR founder lines (Chakraborty et al, 2019) (126 insertions; see Methods) (Fig. 6C; Appendix Fig. S19A). Most *rover* insertions carried a defective *env-F* ORF but had intact *gag* and *pol* genes, classifying them as potential retroelements. These insertions clustered into two distinct variants based on their LTRs measuring 376 bp (32 insertions) or 367 bp (35 insertions) in length. The remaining insertions contained an intact *env-F* ORF, classifying them as potential retroviruses, and exhibited LTRs distinct from the retroelement variants, with lengths of

459 bp (16 insertions), 388 bp (8 insertions), 408 bp (28 insertions), or 315 bp (19 insertions). Thus, over the course of evolution, *rover* diversified into six natural subtypes, all of which are present in most of the analyzed natural isolates.

To investigate the structural divergence among *rover* variants, we performed a comparative sequence analysis of six representative sequences. Overall, the coding sequences of all *rover* variants are highly conserved. The *gag* and *pol* ORFs are nearly identical across variants, with no polymorphisms correlating with the functional status of *env-F* (Appendix Fig. S19B). Notably, *env-F* shared a common mutation in the LTR-367 and LTR-376 retroelement variants: an adenine insertion at position 990 (amino acid 245), resulting in a frameshift followed by a premature stop codon. This suggests that both retroelement variants originated from one initial *env-F* inactivating mutation, and that the LTR-367 variant acquired subsequently an additional 73 bp deletion at the 5′ end of *env-F* (Appendix Figs. S5 and S6B). In contrast to the conserved protein-coding sequences, the non-coding regions, LTR and 5′ UTR, show considerable divergence (Fig. 6D; Appendix Fig. S19B). Numerous single-nucleotide polymorphisms and small insertions and deletions are found within the LTR portion, particularly in the highly variable U3 region. Sequence alignments grouped the LTRs into two clusters: retroviral variants (LTR-388, LTR-408, LTR-459) and retroelement variants (LTR-367, LTR-376), with the structurally defective LTR-315 variant clustering with the retroelements despite containing an intact *env-F* gene (Appendix Figs. S19C and S20). Within the 5′ UTR, *rover* variants differ mainly due to insertions and deletions, with the most notable being a 151 bp tandem repeat expansion, which is present once in the retroviral LTR-388, LTR-408, and LTR-459 variants, three times in the LTR-315 and LTR-376 variants, and four times in the LTR-367 variant (Fig. 6D; Appendix Fig. S20).

To assess the functional significance of variation in *rover* non-coding sequences, we generated lacZ reporter transgenes encompassing the LTR and 5′ UTRs of all six variants and tested their activity in ovaries with defective piRNA pathways (Fig. 6E; Appendix Fig. S21). This analysis revealed a strong correlation between expression patterns and *env-F* status: reporters for retroviral variants (LTR-459, LTR-408, LTR-388) were expressed in follicle cells of soma knockdown ovaries, whereas reporters for retroelement variants (LTR-376, LTR-367) were expressed only in nurse cells of germline knockdown ovaries. The reporter for the inactive LTR-315 variant was not expressed in either condition. The *cis*-regulatory sequences of *rover* variants thus reflect the evolutionary transition from soma-expressed retroviruses to germline-expressed, non-infectious retroelements.

Based on our findings, *rover* represents an ideal model for studying the retrovirus to retroelement transition. Pairwise LTR

divergence analysis suggested that *rover* insertions in DSPR genomes date back several hundred thousand years (Appendix Fig. S21B). Retroviral variants (LTR-315, LTR-388, LTR-459, LTR-408) were generally older than the retroelement ones (LTR-367 and LTR-376), most of which show no pairwise LTR divergence. Consistent with these estimates, only three retroviral LTR-408 insertions with no LTR divergence remain active retroviruses with intact *gag*, *pol*, and *env-F* ORFs, while most retroelement insertions retain intact *gag* and *pol* ORFs, attesting their ongoing activity (Fig. 6C; Appendix Fig. S21B). These observations support the general notion that retroelements derived from ancestral retroviral iERV lineages. Notably, across eleven DSPR genomes, nineteen *rover* insertions reside within the *flamenco* piRNA cluster. These insertions represent at least three independent events that are supported by multiple *rover* insertions in different DSPR strains. All *flamenco* insertions represent retroviral LTR-388 and LTR-408 variants, and all are antisense to the transcription direction of the piRNA cluster. Their pairwise LTR divergence suggests that somatic piRNA-mediated control of the *rover* retrovirus dates to up to 250,000 years ago (average = 144,000 years; *n* = 8) and has been maintained in natural populations around the world (Appendix Fig. S21B). In contrast, the germline piRNA pathway utilizes a stand-alone insertion of a retroelement variant (LTR-367) with no pairwise LTR divergence as Rhino-dependent piRNA source locus (Appendix Fig. 18C). Thus, host silencing against *rover* in soma and germline correlates with the proposed evolution of *rover* variation.

We finally asked whether *rover* retroelement variants evolved from a retroviral ancestor within the *D. melanogaster* host or originated in another species before spreading in *melanogaster* populations. As we could not find *rover* sequences in other *Drosophila* species that closely resembled the *melanogaster* variants, we considered that LTR retroviruses and retroelements are pseudo-diploid, packaging two RNA genomes into a single capsid. Before integration, reverse transcriptase (RT) converts this RNA into double-stranded DNA, switching strands twice, usually producing proviral insertions with identical LTRs at both ends (Negroni and Buc, 2001). If two distinct retroviral variants replicate in the same cell, however, different genomic RNAs can be co-packaged into a single virion, allowing significant genetic exchange through RT-dependent recombination events (Hu and Temin, 1990). Such recombination has been observed in yeast, *Arabidopsis*, and humans (Cappy et al, 2017; Sanchez et al, 2017). In our analysis of all *rover* insertions from DSPR strains, we identified eleven insertions with mismatched 5′ and 3′ LTRs, suggesting they arose from recombination between distinct *rover* variants (Fig. 6F; Appendix Fig. S22). As some of these insertions were found in fixed positions across multiple DSPR strains, they represent at least five evolutionarily independent events. These unique insertions likely resulted from RT strand switching within pseudo-diploid capsids. Specifically, we found recombination events between retroviral *rover* variants (e.g., LTR-408/LTR-459 and LTR-388/LTR-408 recombinants in *flamenco*), between retroelement variants (LTR-367/LTR-376), and even between retroviral and retroelement variants (LTR-376/LTR-408 and LTR-367/LTR-408). Further analysis of the abundant *rover* LTR retroelement insertions, LTR-376 and LTR-367, revealed double recombination events, including the exchange of internal sequences such as *env-F* mutations (Appendix Fig. S23). Collectively, these findings provide strong evidence that active retroviral and retroelement *rover* variants coexisted during *Drosophila melanogaster* evolution. Moreover, recombination among variants may have facilitated the co-evolution of infectivity loss and the expression shift from soma to germline, with initial variants carrying only one of these two sequence changes (Cappy et al, 2017; Hu and Temin, 1990; Sanchez et al, 2017).

## Discussion

This study provides insights into the co-evolution of a whole group of endogenous retroviruses, iERVs, with their host, showing how 'ecological principles' known from evolutionary theory (Brookfield, 2005; Leonardo and Nuzhdin, 2002; Venner et al, 2009) can help explain the complex relationship between transposable elements (TEs) and their host genomes. Using a combination of comparative and experimental approaches, we find that infectious iERVs occupy distinct somatic niches, while non-infectious, derived iERVs are expressed in the germline. Although stochastic events may play a role, we propose that these divergent expression patterns have arisen primarily through adaptive processes. As such, our work provides unique insights into the evolutionary dynamics that underlie the diversification of iERVs and their persistence within the complex "ecosystem" of an animal gonad.

During oogenesis, primordial germ cells and differentiating germline cells interact closely with various somatic cell types. iERVs have uniquely adapted to this complex 'ecosystem', with infectious retroviruses and derived, non-infectious retroelements having evolved different strategies to target the immortal germline genome. Considering that both infectious and non-infectious iERVs can successfully generate new germline insertions (Barckmann et al, 2018; Wang et al, 2018), our results show that two key traits of iERVs, their tissue-specific expression pattern and their *env-F* status, are strictly correlated with their replication strategy.

The acquisition of *env-F* was a key event in iERV evolution, as it allowed LTR retrotransposons to acquire new germline genome integrations without the need for direct germline-expression. Together with matching *cis*-regulatory sequences, this event thereby enabled the ancestral retrovirus to exploit a new niche: the ovarian soma, a previously unprotected habitat with diverse cell types. We argue that the initial gain (and the subsequent secondary losses) of infectivity promoted the diversification within the iERVs, consistent with them being the most diverse LTR retrotransposon clade in *D. melanogaster* and its close relatives. Compared to the ovarian germline, which comprises only two transcriptionally distinct cell types (germline stem cells and differentiating nurse cells), the ovarian soma contains at least eight cell types with distinct gene expression profiles (Hinnant et al, 2020; Slaidina et al, 2021). This greater diversity of cell types may have contributed to retroviruses diverging more readily compared to their directly related, germline-expressed retroelement descendants (Fig. 1A). In addition, competition for host resources or inter-viral interference (Mura et al, 2004; Rosales Gerpe et al, 2019; Sommerfelt and Weiss, 1990; Trono et al, 1989) may have further driven the specialization of iERV expression patterns in distinct niches (Appendix Fig. S24).

Our findings also highlight the impact of the co-evolving piRNA pathway on iERV evolution. Given that this genome defense system suppresses transposon expression, iERV diversification likely

occurred primarily during periods of unchecked replication. Once an iERV lineage came under piRNA control, further diversification would require a temporary loss of this control. Such loss could result from the deletion of iERV sequences in a piRNA cluster (Duc et al, 2019), horizontal transfer of an iERV lineage to a naïve host population or species (Bargues and Lerat, 2017; Scarpa et al, 2024; Schwarz et al, 2021) or shifts in iERV strategy from a soma-expressed virus to a germline-expressed retroelement. We note that endogenous siRNAs have been implicated in transposon suppression in somatic cells (Czech et al, 2008). However, the role of siRNAs in controlling transposons in ovaries appears limited compared to the dominant influence of the piRNA pathway (Barckmann et al, 2018; Pelisson et al, 2007), suggesting that siRNAs are unlikely to impact the evolution of endogenous retroviruses.

Notably, the transition from *env-F*-encoding retrovirus to non-infectious LTR retroelement occurred at least eight times within the iERV clade (Fig. 1D). The resulting retroelement lineages lost somatic expression and adopted germline-specific expression through changes in their *cis*-regulatory sequences, strictly correlated with the loss of functional *env-F* status. This loss may have been passive, as *env-F* is not necessary for germline replication, or actively selected for, since the expression of a membrane fusogen could disrupt germline architecture.

While reconstructing the evolutionary events underlying the soma-to-germline transition is challenging—given the horizontal transfer of iERVs among host species—our analysis of the *rover* lineage, which has no known horizontal transfers, offers unique insights. We identified two broad *rover* subtypes in *D. melanogaster*: retroviral and retroelement variants. These subtypes differ in their *env-F* status and tissue-specific expression, which is determined by sequence changes in their non-coding, *cis*-regulatory LTR and 5′ UTR regions. The loss of functional *env-F* in *rover* variants correlates with a shift from soma to germline expression, suggesting that these variants recapitulate the evolutionary transition from an infectious retrovirus to a germline-restricted retroelement. One of our key findings is that we identified recombination events between retroviral and retroelement variants in natural DSPR strains, suggesting co-existence of both forms during a period of unchecked activity within a single host. These recombination events, likely driven by strand transfers during reverse transcription (Hu and Temin, 1990; Jordan and McDonald, 1999; Sanchez et al, 2017; Temin, 1993), would allow for a scenario where mutations in *cis*-regulatory regions and in *env-F* arose independently, and a recombination event between these mutants gave rise to a functional, non-infectious retroelement with germline-specific expression. Our findings therefore strongly support the evolutionary diversification of the *rover* lineage within the *D. melanogaster* host.

More broadly, our study has significant implications for understanding LTR retrotransposon evolution. First, recent evidence indicates that iERVs or Errantiviruses are also found in many non-insect invertebrate host genomes and share a common origin with lokivirus ERVs in amphibia (Chary and Hayashi, 2024). Second, niche spezialization, which suggests adaptive evolutionary processes driven by competition, is not exclusive to iERVs. For instance, in *Saccharomyces cerevisiae*, the LTR retrotransposons *Ty1* and *Ty3* have partitioned the single-celled host into discrete retro-transposition niches, with *Ty3* expressed in haploid cells and *Ty1* in diploid cells (Sandmeyer et al, 2015). In vertebrates, infectious retroviruses of the *Retroviridae* have co-evolved with their hosts for millions of years. Due to stochastic germline infections, the *Retroviridae* have given rise to multiple lineages of endogenous retroviruses (ERVs) in vertebrate genomes (Johnson, 2019). Notably, several vertebrate ERVs are expressed during specific developmental time windows. For example, during human preimplantation development, different HERV lineages display striking, often non-overlapping spatiotemporal patterns of LTR activity (Carter et al, 2022; Goke et al, 2015). Third, some vertebrate ERVs also display a loss of their *env* gene. An example is the intracisternal A particle (IAP) in mice, a highly active ERV lineage, which evolved from a retroviral form, IAP-E, that encoded a functional *env* gene (Ribet et al, 2008). Similar to non-infectious iERVs in *Drosophila*, IAPs are expressed in the mouse germline (Dupressoir and Heidmann, 1996) and replicate successfully despite being targeted by the piRNA pathway (Carmell et al, 2007; Magiorkinis et al, 2012). This suggests that the ecological and evolutionary principles driving iERV evolution in *Drosophila* could similarly influence the evolution and adaptation of vertebrate ERVs. As it is common to all LTR-retroviruses and -retroelements, this may include mechanisms such as reverse transcriptase-mediated mutagenesis and recombination. In this context, our findings provide critical insights into the origins of eukaryotic *cis*-regulatory elements, which have repeatedly been co-opted from LTR regulatory sequences (Chuong et al, 2017).

# Methods

**Reagents and tools table**

| Reagent/Resource | Reference or Source | Identifier or Catalog Number |
|---|---|---|
| **Experimental models** | | |
| Drosophila fly lines | | Dataset EV1 |
| **Recombinant DNA** | | |
| **Antibodies** | | |
| Rabbit polyclonal Anti-Piwi | (Mohn et al, 2014) | |
| Mouse monoclonal anti-Piwi 8C2-E4 | (Senti et al, 2015) | |
| Mouse monoclonal anti-Aub 8A9-D7 | (Senti et al, 2015) | |
| Mouse monoclonal anti-Ago3 7B4-C2 | (Senti et al, 2015) | |
| Mouse monoclonal anti-Armadillo N2 7A1 | DSHB | |
| Promega anti-Beta-Gal | Promega | Z3781 |
| Rabbit polyclonal anti-Rover Gag | (Senti et al, 2015) | |
| Rabbit polyclonal anti-McClintock Gag | This study | |
| Rabbit polyclonal anti-Gypsy Gag | This study | |
| **Oligonucleotides and other sequence-based reagents** | | |
| Primers for reporter constructs | | Dataset EV1 |

| Reagent/Resource | Reference or Source | Identifier or Catalog Number |
|---|---|---|
| Primers for env splice junctions | | Dataset EV1 |
| smFISH probes | | Dataset EV1 |
| **Chemicals, Enzymes and other reagents** | | |
| Hoechst 33342 | Thermo Fisher Scientific | 62249 |
| Spider beta-Gal | Dojndo | SG02 |
| **Software** | | |
| Dendroscope (3.8.10) | (Huson and Scornavacca, 2012) | |
| Figtree (1.4.3) | http://tree.bio.ed.ac.uk/software/figtree/ | |
| MASCE (2.0) | (Ranwez et al, 2018) | |
| IQTREE | (Trifinopoulos et al, 2016) | |
| Ancestral character states (phytools) | (Revell, 2024) | |
| BiSSE (diversitree) | (FitzJohn et al, 2009) | |
| RaxML version8 | (Stamatakis, 2014) | |
| MrBayes 3.2 | (Ronquist et al, 2012) | |
| Custom script to calculate piRNA cluster silencing potential | This study | https://gitlab.com/BrenneckeLab/cluster-silencing-potential-analysis |
| Graph Pad Prism version 10 | | |

## Annotation of LTR retrotransposon consensus sequences

Consensus sequences for all *Metaviridae* and *Belpaoviridae* LTR retrotransposons in *Drosophila melanogaster* were collected from the BDGP and Repbase databases as well as from primary literature (Bao et al, 2015; Kaminker et al, 2002; Kapitonov and Jurka, 2008; Senti et al, 2015). In most cases, existing annotations for LTR, 5′ UTR, and coding sequences of *gag*, *pol*, and *env-F* were adopted. For several retrotransposons, we manually curated the consensus sequences. For this, existing consensus sequences were used to search the dm6 reference genome for the longest and most complete insertions of the respective lineage. Within these, we identified sequence stretches representing open reading frames for *gag*, *pol*, and *env-F*, and aligned those with MASCE (Ranwez et al, 2018) to all other *gypsy/gypsy* clade sequences for the same ORF. This allowed the correction of isolated SNPs or small INDELs in *env-F* for the following consensus sequences: *gtwin*, *gypsy5*, *rover*, *HMS Beagle2*. For alignments containing generally incomplete ORFs (inactive lineages), additional shorter fragments were identified in dm6 and used to conservatively correct and fill gaps, frame shifts, and stop codons, always using MACSE alignments. This process was reiterated until the respective consensus sequence could not be improved further using additional fragments. The curated ORF sequences were used to assemble

full-length consensus sequences that were used in this study. The following consensus sequences were curated for multiple ORFs: *accord2*, *gypsy2*, *gypsy3*, *gypsy7*, *gypsy9*, *gypsy10*. To annotate the complete *env-F* coding sequences, splice junctions of sub-genomic *env-F* transcripts were determined. Total RNA was extracted from ovaries of *traffic jam*-GAL4 UAS-*vreteno*[GD] females (fly stocks listed in Dataset EV1), PCR products were amplified using random primed first strand cDNA and primers flanking known and anticipated *env-F* splice junctions (primer sequences listed in Dataset EV1). PCR amplicons were subcloned and sequenced. From the resulting full-length Env-F protein sequences, signal peptide and transmembrane domains were predicted with signalP 5.0 and target 1.1 or TMHMM 2.0, respectively (Almagro Armenteros et al, 2019a; Almagro Armenteros et al, 2019b; Krogh et al, 2001). All consensus sequences used in the study are provided in Dataset EV2. Based on the curated set of consensus sequences, we determined which lineages were represented by at least one structurally intact (active lineage) or exclusively structurally defective copies (inactive lineage) in the dm6 reference genome. For all retro-element consensus sequences lacking full-length *env-F*, the sequence from the end of *pol* extending into the 3′ LTR were aligned to full-length *env-F* sequences from the same subclade using MASCE to identify *env-F* ORF fragments.

## Sequence alignments and phylogenetic analyses

The open reading frames of *gag*, *pol* and *env-F* in the curated consensus sequences were aligned using MACSE2.0 (Ranwez et al, 2018). Sequences for *gag* were restricted to the central core domain, sequences for *pol* to the Protease domain until the end of the conserved Integrase domain, sequences for *env-F* were trimmed at the N-terminus until the beginning of the signal peptide (*idefix* subclade). The resulting MACSE alignments were used to estimate phylogenetic trees using three methods, RAxML and IQ-tree (Stamatakis, 2014; Trifinopoulos et al, 2016). For RAxML, the best-scoring substitution model was determined among the amino acid models as LG + F + Γ (LG with empirical base frequencies and the Γ model of rate heterogeneity (Le and Gascuel, 2008), and clade support values were calculated from non-parametric bootstrapping implemented in RAxML based on 1000 replicates. The phylogenetic relationships were confirmed by performing phylogenetic analysis with and without the outgroups in IQ-TREE with 1000 ultrafast bootstraps. Phylogenetic trees were visualized and analyzed using Dendroscope (Huson and Scornavacca, 2012). Diversitree (FitzJohn et al, 2009) was used for the statistical analysis of the ancestral character state using the RAxML Pol tree without the *gypsy/mdg3* outgroup and states of either functional *env-F* or clear *env-F* fragments versus those lacking clear env-F evidence (*gypsy7* and *transpac*) at the terminal branches. BiSSE was used to estimate if speciation and extinction rates of retroviruses (with functional *env-F*) and retro-elements (without functional *env-F*) differed. LTR alignments were performed using MUSCLE alignments in MegA-lign (DNASTAR) or the web interface at www.ebi.ac.at (Edgar, 2004). The approximate ages of individual LTR-retrotransposon insertions in dm6 by pairwise LTR comparisons were estimated using the formula $T = K/2xr$ (Bowen and McDonald, 2001) with $r = 0.0346$, assuming ten generations per year (Cutter, 2008; Keightley et al, 2009; Obbard et al, 2012). The nucleotide sequence divergence of individual iERVs within their respective subclades was assessed by extracting LTR, 5′ UTR, and ORFs for *gag* (core),

*pol* (RT-RNAseH domain), and *env-F* from the respective consensus sequences, aligning them with MUSCLE (for LTR) or MASCE (for ORFs), and determining the percent sequence identities and the percentage of aligned nucleotides in Clustal Omega (Sievers et al, 2011). To detect remnant *env-F* ORFs in retroelements, the sequence interval from the 3′ end of *pol* and to the including downstream LTR was extracted and aligned to the *env-F* ORFs of retroviruses in the same subclade using MACSE2.0.

## Genetic analyses

All fly stocks used in this study are listed in Dataset EV1. Tissue-specific knockdowns of piRNA pathway genes in the ovarian soma or germline were performed as described previously (Dietzl et al, 2007; Handler et al, 2013; Ni et al, 2011; Senti et al, 2015). For knockdown experiments in the male germline (testes), we used a driver line combining NGT40 (2nd chromosome) and *bam*-GAL4:VP16 (3rd chromosome) (Dobbelaere et al, 2023) with shRNA-lines targeting *white* (control) or *aub+ago3*. To identify somatic cell types, GFP-trap lines (Buszczak et al, 2007; Morin et al, 2001) for *CG14207* (CB02069), *Drat* (CB03410), *fax* (CC01359) and *fas3* (G00258), labeling terminal filament cells, cap cells, escort cells or germarium early follicle cells and polar cells, respectively, were combined with the *tj*-Gal4 driven knockdown system. For germline knockdowns, a *GFP-nup107* transgene was introduced for labeling the nuclear envelope (Senti et al, 2015).

## polyA RNA-seq

Total RNA was extracted with TRIzol from ovaries of adult flies or from dechorionated and washed 0–1 h old embryos (prior to the onset of zygotic transcription). After RNAeasy column purification with on-column DNAse I digest, eluted RNA was 2x polyA selected using magnetic oligo-dT beads and further processed as strand-specific RNA-seq libraries (Zhang et al, 2012). All RNA-seq libraries are listed in Dataset EV1.

## smFISH probes

smFISH probes were designed as previously described (Senti et al, 2015) and ordered either directly from LGC/Stellaris or as DNA oligos that were subsequently labeled in-house (Gaspar et al, 2017). All probe sets are listed in Dataset EV1. lacZ HCR probes were designed according to (Glotzer et al, 2022).

## Antibodies

All antibodies used in this study are listed in Dataset EV1. For newly generated transposon antibodies, we selected antigens that show a high level of sequence difference within an iERV subclade (*McClintock* Gag: ITEAQTAENFRPQASEQANS; *gypsy* Gag: EAPKQKDPKEEYEKTAKAA). Peptides were synthesized with terminal cysteine residues and coupled to KLH. Polyclonal antisera were raised in rabbits using the rapid protocol (Eurogentec).

## smFISH, HCR, immuno-fluorescence, and ß-Gal stainings

Ovary and testes single-molecule fluorescent RNA in situ hybridization (smFISH) and antibody stainings in ovaries or embryos were

performed as previously described (Senti et al, 2015). For smFISH/antibody co-stainings, we first performed RNA smFISH, followed by anti-Armadillo stainings in 2xSSC-based buffers. RNA smFISH stainings in early embryos were according to (Trcek et al, 2017). HCR was performed as in (Glotzer et al, 2022). Reporter transgene expressions were analyzed by *lacZ* RNA smFISH, *lacZ* HCR-FISH, or anti-β-Gal stainings. Chromogenic β-Galactosidase assays were performed with blue precipitate-forming X-Gal (Handler et al, 2013) or fluorescent SPIDER β-Gal substrate: Ovaries of lacZ reporter-expressing flies were fixed with 4% PFA in PBS at room temperature and stained in PBS containing 1 μg/ml Hoechst 33342 and 2 μM fluorescent SPiDER β-gal reagent at 37 °C for 30 min, followed by 3x PBS washes and mounting in Prolong Diamond mounting medium (Doura et al, 2016). All specimens were imaged using a Zeiss LSM780 confocal microscope or on an Axio Imager.Z2 widefield microscope with an Axiocam 506 color camera for X-gal stainings.

## LacZ reporter transgenes and functional analysis

Transposon reporters were based on a vector backbone with attB, mini-white marker, multi cloning site, and a 3xHA-lacZ reporter followed by a 3′ UTR containing a short *ftz*-intron, and *p10* terminator. *cis*-regulatory sequences of individual iERVs were amplified from genomic BACR clones (Hoskins et al, 2000) or Pacman clones (Venken et al, 2009) containing a corresponding full-length iERV insertion. PCR amplicons included the entire 5′ LTR and the 5′ UTR plus the *gag* ATG. Sequence-verified reporter constructs were integrated into *attP40* (2nd chromosome) (Markstein et al, 2008). Obtained reporter transgenes were combined with UAS transgenes for control or piRNA targeting RNAi constructs and crossed to the different Gal4 driver lines. The *flamenco* and *cluster 77B* transcriptional reporter constructs were generated similarly (using primers listed in Dataset EV1) and were integrated into *attP2* (*flamenco*) and *attP40* (*cluster 77B*).

## piRNA cluster silencing potential

Genomic sequences (dm6) from of the uni-strand piRNA clusters *flamenco* and *cluster 77B* and from the dual-strand piRNA clusters *38C*, *42AB*, and *80F* were parsed into all possible 25mer sequences, shifted by 1-nt, to mimic piRNAs. These 25mer sequences were mapped to all iERV consensus sequences with no mismatches (similar results were obtained with three mismatches). The mappings along each consensus sequence were recorded in a binary fashion, resulting in the percentage of each iERV sequence that was complementary to cluster 25mers. Chromosomal coordinates (dm6) for the analyzed piRNA clusters are listed in Dataset EV1. The original code is available at https://gitlab.com/BrenneckeLab/cluster-silencing-potential-analysis.

## *cluster 77B* mutagenesis

The presumptive promoter element of *cluster 77B* was deleted using CRISPR/HDR with two guide RNAs (GCTATGTAACGCCGTCTGCA and GCACAGCAAAATTTGCACTG) that were cloned into pCFD4D (Ge et al, 2016). The HDR repair template consisted of homology arms (784 bp left arm and 1022 bp right arm) flanking a GMR-*white+* marker that itself was flanked by FRT sites. Both plasmids were injected into

embryos laid by *y,w; actin-Cas9* flies. Mutant alleles were sequence verified. The *FRT-GMR-white⁺-FRT* cassette for *77B^{m1-1}* was removed by crossing to a hs-Flp; Dr/TM3 strain and heat shocking. The Minos allele inside *cluster 77B* is *Mi(MIC)MI07908* and was obtained from the Bloomington stock center. All mutant alleles were crossed out for at least two generations to virgins of the *w; Dr/TM3,Sb* strain to homogenize their genetic background prior to phenotypic analysis.

## Small RNA-seq libraries

Small RNA-seq libraries from ovaries were generated as previously described (Baumgartner et al, 2022), taking advantage of the Argonaute isolation protocol via TraPR (Grentzinger et al, 2020). All libraries are listed in Dataset EV1.

## Small RNA-seq analysis

Small RNA-seq and PIWI protein IP small RNA-seq libraries were analyzed as previously (Senti et al, 2015), but using the updated iERV consensus sequences (no mismatches for mappings). Total small RNA-seq libraries analyzed were from ovaries of *MTD>rhino-sh* versus *MTD>white-sh* (control) (Mohn et al, 2014). The piRNA analysis for *cluster 77B* mutants used genome-unique piRNAs mapping to piRNA *cluster 77B* with no mismatches. Tissue-specific PIWI protein-IP-small RNA-seq libraries were analyzed as in (Senti et al, 2015).

## Comparative analysis of piRNA *cluster 77B* in other species and *D. melanogaster* isolates

We analyzed the genomic organization of *cluster 77B* in *D. melanogaster* iso-1 (dm6), *D. simulans,* and *D. sechellia* (Drosophila 12 Genomes C et al, 2007). For the strain analysis, we used assembled genomes of the starting strains of the *Drosophila* synthetic population resource (DSPR) (Chakraborty et al, 2018) and annotated the content of the *cluster 77B* locus using a repeat masker analysis (Bao et al, 2015). The insertion ages of contained LTR-retrotransposons were calculated as described above.

## Comparative *rover* insertions analysis

*rover* insertions in DSPR strains were identified by repeat masker analysis and extracted as individual sequences from each genome including ~500 bp *Drosophila melanogaster* unique genomic 5′ and 3′ of the *rover* sequence. Each *rover* insertions sequence file was annotated for intact Gag, Pol and Env open reading frames and its LTR sequences using DNAstar or SNAPgene and variant numbers were recorded and compiled by DSPR genome and location therein. *rover* insertions in *flamenco* were identified and by genomic position relative to known *flamenco* annotations. Assignment to the different *rover* variants by LTR size was done using the 5′LTR. Alignments of *rover* sequences (full length or separate parts, LTR, 5′UTR or open reading frames or its domains) were performed by MUSCLE or MASCE 2.0. *rover cis*-regulatory sequences frequency plot calculated using an LTR-5′UTR multiple sequences alignment of six representative *rover* variants. From this alignment, we calculated the frequencies per site in the alignment that contain aligning sequences (indicating insertions and deletions between

*rover* variants) and the frequency of the identical, or variant SNP in the alignment ranked by frequency.

## *rover* insertion sequence recombination plots

*rover* variant/type reference sequence variation was calculated by comparing two insertions of a known type (e.g., two insertions of the LTR 408 = parental type1) to two insertions of another *rover* type (e.g., two insertions of the LTR459 = parental type2). The only exception was for the LTR388 variant/type for which only one good complete sequence exists (from *iso-1*). For each recombination plot, the parental *rover* reference sequences and identified or potential recombinant *rover* insertions sequences were aligned in MUSCLE. Only the sequence variants (SNPs and INDELs) present in both parental insertion sequences of one *rover* variant/type or the other *rover* variant/type were used as reference sequence variants. All other variations present in recombinant *rover* insertions were called as non-reference variants. For each position in the alignment only identified parental or non-parental variations were plotted. For structural reference to each recombination plot alignment, *rover* functional sites were mapped to the respective alignment and are displayed as a model at the bottom of each plot.

## Data availability

All fly stocks generated for this study are available from VDRC, antibodies are available upon request, NGS data are available from GEO (GSE239985), Image Raw Data is available at BioImage Archive (S-BIAD1883).

The source data of this paper are collected in the following database record: biostudies:S-SCDT-10_1038-S44318-025-00471-8.

## Peer review information

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

## Acknowledgements

We thank the VBCF core facilities (NGS, VDRC) for sequencing and fly stocks, the in-house Protein Chemistry and BioOptics facility for excellent support, and the Fly&Worm Facility for transgenesis and CRISPR-mediated genome engineering. We thank Thomas Harivel, Matthias Schäfer, and Maria Novatchkova for experimental and analytical support, Chantal Vaury, Jereoen Dobblare, the Bloomington and VDRC stock centers for flies, and the Developmental Studies Hybridoma Bank (DSHB) for antibodies. We thank members of the Institute of Population Genetics (VetMed University Vienna), the Brennecke laboratory, and Andrea Betancourt for discussions, and Life Science Editors, Andrea Pauli, Ortrun Mittelsten Scheid, Alejandro Burga, Arturo Marí-Ordóñez, Alexander Hayward, and Justin Blumenstiel for comments on the manuscript. This work was funded by the Vienna Science and Technology Fund WWTF (grant 10.47379/MA16061; CK), the Biotechnology and Biological Sciences Research Council BBSRC (grant BB/W000768/1; CK), the ERC (ArchAdapt; CS), the Austrian Academy of Sciences (JB), and the Austrian Science Fund FWF grants (P32935; CS) and (P33715-B; KAS). BR is supported by a DOC Fellowship from the Austrian Academy of Sciences.

## Author contributions

**Kirsten-André Senti**: Conceptualization; Data curation; Formal analysis; Supervision; Funding acquisition; Validation; Investigation; Visualization; Methodology; Writing—original draft; Writing—review and editing. **Baptiste Rafanel**: Investigation; Visualization; Methodology; Writing—review and editing. **Dominik Handler**: Software; Formal analysis; Investigation; Methodology; Writing—review and editing. **Carolin Kosiol**: Formal analysis; Funding acquisition; Investigation; Writing—review and editing. **Christian Schlöetterer**: Resources; Funding acquisition; Writing—review and editing. **Julius Brennecke**: Conceptualization; Resources; Supervision; Funding acquisition; Visualization; Writing—original draft; Project administration; Writing—review and editing.

Source data underlying figure panels in this paper may have individual authorship assigned. Where available, figure panel/source data authorship is listed in the following database record: biostudies:S-SCDT-10_1038-S44318-025-00471-8.

## Disclosure and competing interests statement

The authors declare no competing interests.

