## [Peer Review File · The EMBO Journal]

Co-evolving infectivity and expression patterns drive the diversification of endogenous retroviruses

Kirsten-André Senti, Baptiste Rafanel, Dominik Handler, Carolin Kosiol, Christian Schloetterer, and Julius Brennecke

Corresponding author: Julius Brennecke (julius.brennecke@imba.oeaw.ac.at)

Review Timeline:

SubmissionDate:	29th Nov 24
Editorial Decision:	23rd Jan 25
Revision Received:	29th Apr 25
Accepted:	15th May 25

Editor: Yehu Moran

Transaction Report:

Dear Dr. Brennecke,

Thank you for submitting your manuscript for consideration by the EMBO Journal. It has now been seen by three referees whose comments are enclosed. As you will see, all three referees express great interest in your manuscript and are in favor of publication, pending satisfactory minor revision.

Given the referees' highly positive recommendations, I would like to invite you to submit a revised version of the manuscript, addressing the comments of all three reviewers. I should add that it is EMBO Journal policy to allow only a single round of revision, and formal acceptance of your manuscript will therefore depend on the completeness of your responses in this revised version.

In addition to the referees' comments please see below the comments by our editorial office team and data integrity analysts. It is very important that you handle carefully all these comments and relate to them in detail in your revision.

We generally allow three months as standard revision time. As a matter of policy, competing manuscripts published during this period will not negatively impact on our assessment of the conceptual advance presented by your study. However, we request that you contact the editor as soon as possible upon publication of any related work, to make us aware of such matter.

Thank you for the opportunity to consider your work for publication. I very much look forward to your revision.

Sincerely,

Yehu Moran
Academic Editor
The EMBO Journal

We realize that it is difficult to revise to a specific deadline. In the interest of protecting the conceptual advance provided by the work, we recommend a revision within 3 months (23rd Apr 2025). Please discuss the revision progress ahead of this time with the editor if you require more time to complete the revisions.

Specific comments by editorial assistant:

- *AUTHOR CHECKLIST: missing. Please include.
- *PBP: missing. please include
- *Figures in separate files: main figures should be uploaded as individual, high resolution figure files, the legends should be compiled at the end of the manuscript text.
- *Keywords: missing. Please complete.
- *AUTHORS: there is a discrepancy in the first authors' name: Kirsten-André Sent (in the manuscript), Kirsten Andre Sent (in our system); please confirm which is correct.
- *ORCID ID: missing for co-corresponding author Senti. Our editorial assistant sent an email with a link.
- *DATA AVAILABILITY SECTION: included but should be renamed
- *FUNDING: please merge with Acknowledgements.
- *Author Contributions: Please remove from the manuscript text; these should be in our system only
- *Disclosure and competing interests statement: please rename
- *DATASET EV LEGENDS: Is the text file labeled "Document S1" computer code? In that case it should be renamed "Computer Code EV1". If not it should be made an EV Dataset. The excel table uploaded as a reagents and tools table, labeled "Table S1", should be renamed "Dataset EV1".
- *APPENDIX 1 FILE WITH ToC: the Methods (with the Reference list) should be removed from the supplementary information file and added to the manuscript text. The 24 supplementary figures should be renamed "Appendix Figure S1" - S4. Please add page numbers to the table of contents and rename the file "Appendix".
- *SOURCE DATA: Hannah Sonntag sent the email on 22nd January 2025. Please address all comments.
- *REAGENT TABLE: missing, please ask the authors to upload it as a separate file using our template.
- *SYNOPSIS IMAGE: Please provide.
- *SYNOPSIS TEXT: Please provide.

Additional Notes:

- Please correct the order of the manuscript sections; this should be: Abstract, Keywords, Introduction, Results, Discussion, Methods, Acknowledgements, Disclosure and competing interests statement, References, Figure legends, Tables and their legends, Expanded View Figure legends.
- The list of supplementary materials should be removed from the manuscript text.

FIGURE CHECK was performed on Thursday, 23rd of January 2025.

The following issues must be addressed:

Undisclosed image reuse between Figure 2 C,D,E,F,G and Appendix Figure S10 A,D,F,H. Please check if the reuse is justified and clarify with the authors. If reuse is justified it needs to be cited in the Appendix figure legend.

Undisclosed image reuse between Figure 3D and Appendix Figure S10 F. Please follow same process as above.

Very similar features in the highlight section of Appendix Figure S10D and Appendix Figure S11F. One is ZAM and the other is Burdock. Please check and clarify.

Undisclosed image reuse between Appendix Figure S14A Transpac and Appendix Figure S15 Transpac. Please follow same process as above.

Appendix Figure S21 - LTR-459 variant. Lower 2 cells appear the same. This appears to be an irregularity in the figure. Please check and clarify.

- Figure legends:

1. Please indicate what */ **/ ***/ **** represents; if this represents p value(s), please specify the exact p value in the legend(s) of figure(s) 1D
2. Please indicate what */ **/ ***/ **** represents; if this represents p value(s), please indicate the statistical test used and where appropriate, specify the exact p value in the legend(s) of figure(s) 2A, B.
3. Please note that information related to n is missing in the legend of figure 1D.
4. Please note that the scale bar needs to be defined for figures 3B-D; 6E.

5. Please note that scale bar and its definition are missing for figure 2D.

Referee reports:

Referee #1:

This is a remarkable paper providing evidence that insect endogenous retroviruses (iENV, like gypsy and ZAM), which are expressed in somatic cells (follicle cells of the ovary), gave rise to germline-expressed non-infective retroelements (like McClintock, accord and burdock). The authors show that gypsy and ZAM-infected follicle cells produce virus-like particles that infect neighboring oocytes where they integrate into the germline. In addition to establishing the perfect correlation between the presence and expression in iENVs of env-F, the env gene in the gypsy-class elements studied here, and the loss of env-F expression in retroelements, the authors explore the piRNA regulation of elements across this spectrum. Somatic piRNA expression from the flamenco piRNA cluster appears to control most of the retroviruses, but there are no 17.6 piRNAs produced from this cluster, leading to the discovery that the 77B piRNA cluster is also unidirectional and produces piRNAs against 17.6. They go on to show that 77B is necessary for control of 17.6. For me, the most impressive part of the paper centers on the analysis of rover elements. This family of elements spans the range from essentially true retroviruses (with somatic expression, viral particle formation, and other endogenous retroviral properties) to germline-restricted, non-viral particle forming transposable elements. Most impressively, they also document recombination events both within and between the retroviral and the retroelement rover LTRs. In addition to documenting all these features of rover elements, they also demonstrate the host regulation of each class, and they do molecular evolution analysis to show that the transposons are derived from the endogenous retroviruses (complete with a plausible inference of their age).

The authors make solid use of many tools from computational evolutionary genomic analysis, starting with phylogenetics. The phylogenetic tree of the Metaviridae forms the central focus of the analysis, as it contains the five major clades of the gypsy-class elements, all with a gag and pol gene. Seventeen of the lineages on this tree have env-F, and all are in the gypsy-class elements. Eleven lineages lack functional env-F and show instead varying degrees of sequence divergence, reflecting different ages of loss of function. All inactive element insertions of both the retrovirus and retroelement classes are located in pericentromeric heterochromatin, and the high LTR sequence divergence reflects their great age. Overall, the data are consistent with these iERVs being monophyletic within *D. melanogaster*, an interesting finding as it means that the diversification of this element family, spanning both endogenous retroviruses and germline-limited retroelements (transposons), occurred entirely within this species.

The experimental manipulations include RNAi knockdowns of piRNA pathway components, driven with highly tissue-specific drivers. For example, tj-Gal4-driven Zucchini shRNA accomplished knockdown of ovarian somatic piRNAs, and MTD-Gal4 driven Aubergine shRNA accomplished knockdown of piRNA expression in the ovarian germline. Single-molecule FISH provided clear evidence of changes in tissue-specific expression (e.g. that McClintock transcripts were only detected in nurse cells). The Supplementary figures show clearly the extensive differentiation in tissue specificity of the different gypsy-class elements, including targeting of the 8 different somatic cell types in the fly ovary. In particular "retroviruses with strong derepression at the RNA-seq level (ZAM, gypsy, springer, gypsy6, HMS-Beagle2) were expressed in broad domains, while those with moderate derepression (idefix, 17.6, quasimodo, 297, gypsy5) exhibited more restricted expression niches." Moreover, viruses from the idfix subclade were predominantly expressed during early oogenesis, whereas members of the ZAM, springer and Beagle subclades were active mainly at later stages. This suggested that different iERV subclades target the germline genome at distinct stages of oogenesis. These observations suggest that infectious iERV lineages collectively occupy the full spectrum of ovarian somatic cell types, quite convincingly support the notion that they occupy different "niches" in the host.

I conclude with a simple, direct statement - this is the best paper I have read all year!

Small points, all to be considered optional:

1. I notice in the phylogenetic tree of Fig 1 that tirant is fairly closely related to ZAM, but there is no further mention of tyrant. Given the interest in its recent invasion, I was hoping the authors could say whether tyrant the host had developed similar regulation of tyrant as is seen for ZAM.
2. In the reference list, Carmell et al.(2007) is listed as "in press"
3. I am a bit surprised that there is no mention of siRNAs, as others have implicated them in regulation of endogenous retroviruses and somatic transposable elements.
4. The term "niche partitioning" is presented with the implication that there is some sort of competitive exclusion keeping elements from having overlapping niches. The data clearly show the conspicuous level of niche specialization, but there really is not direct evidence of competitive exclusion. This would require showing that the presence of element A in a tissue impedes the success of element B in the same tissue, either by reducing transposition rate, or by reducing host survival or fecundity. It is a

minor point, but I believe that "niche specialization" is a bit more accurate here. Of course it is also a really interesting speculation that the niche specialization came about as a result of active competitive exclusion.

Referee #2:

In this study, the authors systematically analyzed expression of endogenous retroviruses (ERVs) in the *Drosophila* ovary, comparing elements coding an envelope gene (retroviruses) and those that do not (retroelements). They find that the ancestor of the monophyletic group of ERVs likely encoded an envelope gene, which had been lost in certain lineages. They report an intriguing correlation between the presence of the envelope (and hence infectivity) and specific expression in soma versus germline: elements containing an envelope are expressed in the soma (and suppressed by the somatic piRNA pathway), whereas retroelements without an envelope are expressed in the germline (and suppressed by the germline specific piRNA pathway). Strikingly, different retroviruses are expressed in different cell types of the ovary, a process the authors refer to as niche partitioning. Using lacZ reporters, they demonstrate that these differences in expression are driven by (adaptive) changes in the LTR and 5' untranslated regions of the elements. Finally, they report an intriguing case of an ERV lineage (rover) that seems to be in a transitional state from infectious retrovirus to a retroelement.

Overall, the paper provides interesting new insights about transposon evolution, with likely relevance also beyond the *Drosophila* model systems. I found the idea of niche partitioning especially intriguing. The work is very well executed, a pleasure to read, and extensively supported by 24 (!) supplementary figures. I have only minor textual comments and suggestions for consideration.

Panel indicators in Figure 1 do not correspond to the figure.

Figure 1C: The difference between the red boxes indicating env gene in consensus (full-length according to the legend) and the white box under the env gene (intact ORF) for some elements is not clear to me. Please clarify.

Figure 4A: I did not quite understand the explanation of circle size in the legend of panel A. In the figure itself I propose to change similarity with identity.

Figure 6D: the labeling top1, top2, top3 was not very intuitive to me. Consider rephrasing. Font of x-axis labelling is unreadable.

Method, piRNA cluster silencing potential: "piRNA clusters were parsed into all possible 25mer sequences". Was this done with a 1-nt offset or were the sequences parsed head-to-tail?

Supplementary figure 3: delete "from" from the caption title.

Supplementary figure 5: please define triangle and asterisks.

Supplementary figure 7B: the y-axis scale is missing from the right panel. Also labelling is missing for the x-axis.

Supplementary figure 10: panels of the specific knockdowns are also presented in the main figure; I wonder whether the controls are not better presented in the main figure as well. If the authors choose not to, then I suggest at least indicating in the legend that panels are duplicated (likewise for other supplementary figures in which panels are also presented in main text-figures).

Supplementary figure 22, legend: it was unclear to me what "references" was referring to.

Line 110: "active and inactive lineages". How are inactive lineages defined?

Line 183: "ZAM transcripts accumulated in yolk granules". Figure 2D suggests that the transcripts are at the periphery of the granules. The authors may want to comment on this.

Referee #3:

- general summary and opinion about the principal significance of the study, its questions and findings

This very elegant and informative study (Senti et al., manuscript submitted: EMBOJ-2024-119776) offers insights into the co-evolution of insect endogenous retroviruses (iERVs) and their host genomes (*Drosophila*, mainly *melanogaster*), highlighting how ecological principles from evolutionary theory help to explain the complex relationship between transposable elements (TEs) and their hosts. The authors find that infectious iERVs primarily occupy somatic niches in female gonads, while non-

infectious iERVs are expressed in the germline cells of the gonads. The divergent expression patterns of these iERVs are primarily driven by adaptive processes, rather than random events, contributing to their persistence within the gonad peculiar ecosystem.

In particular, the study shows that during oogenesis, iERVs adapt to interact with the ovarian somatic and germline cells, with infectious retroviruses targeting the somatic cells and non-infectious retroelements targeting the germline cells. The key evolutionary event that enabled this shift was the acquisition of the env-F gene, which allowed retroviruses to integrate into the germline genome without directly being expressed in the germline. This event, alongside changes in CIS-regulatory sequences, fostered the diversification of iERVs, making them the most diverse Long Terminal Repeat (LTR) retrotransposon clade in species such as *D. melanogaster*.

The ovarian soma, with its greater diversity of cell types (over 7) compared to the germline (2 cell type), likely contributed to the greater divergence of iERVs compared to their germline-expressed descendants. This ecological niche partitioning may have also been driven by competition for host resources as well as viral interference. Furthermore, this study underscores the role of the piRNA genome defense system in the evolution of iERVs. Periods of unchecked iERV replication likely spurred diversification, which was later controlled by the piRNA system, with further diversification requiring a temporary loss of this control. Finally and importantly, the authors shed light on a newly discovered somatic piRNA cluster (77B) and characterise its role and importance of partial redundancy with flamenco piRNA clusters however at different "geographical" and temporal states of expression. This part is extremely elegant including the numerous functional characterization using multiple lacZ reporter systems (on ERVs and on clusters) to support the comparative sequencing analysis discoveries and hypotheses. This part also includes an elegant cluster sequence evolution comparison (including mobile element insertion) with 2 other *Drosophila* species.

This study also identifies at least 8 iERV lineages (Rover) transitioning from infectious retroviruses (iERV) to non-infectious retroelements, often associated with the loss of functional env-F. These retroelements switched from somatic to germline-specific expression through changes in CIS-regulatory sequences, potentially to avoid disrupting the germline's structure. The study highlights how recombination between retroviral and retroelement variants in natural strains of *D. melanogaster* supports the hypothesis that such recombination events drove the transition from infectious retrovirus to germline-restricted retroelement.

Broader implications of the authors findings include the evolutionary patterns shared by iERVs in insects and endogenous retroviruses in other species, such as non-insect invertebrates and vertebrates. In vertebrates, similar ecological principles and evolutionary processes may apply to ERVs, as shown by the loss of the env gene in certain ERV lineages, such as the intracisternal A particles (IAPs) in mice, which are expressed in the germline despite being targeted by the piRNA pathway. These findings suggest that mechanisms like reverse transcriptase-mediated mutagenesis and recombination may be central to the evolution of retroviruses across eukaryotic species, providing new insights into the origins and evolution of eukaryotic cis-regulatory elements.

- specific major concerns essential to be addressed to support the conclusions

No major concern, but please check carefully a potential wrong calling (core text and legend) or labeling of the figure panels (Figure 1 mostly).

Starting from p4, lane 107: "To comprehensively characterize the envelope ORFs of Metaviridae, we considered that Env is typically translated from a spliced subgenomic transcript (Figure 1B)". Figure 1B is an alignment not the cartoon of spliced Env (also the legend is not fitting with the letter).

Same lane 118, I think a wrong figure is called: (Figure 1C; Supplementary Figure 3) check if Figure 1B has to be called here.

Same lane 124: phylogenetic architectures (Figure 1D; Supplementary Figure 4A, B) not sure the right figure to call here is Figure 1D (likely Figure 1C).

Then please check before and after if the figures are called accurately, it may be a wrong labeling of the figure panels in figures provided for reviewing. Please check carefully the corresponding legend after correction as well.

- minor concerns that should be addressed

In the introduction, p3 lane 80, permissive flamenco allele is stated, it will help the less specialised readers if authors precise (and cite) what permissive (and restrictive) flamenco alleles are.

Lane 99 p4 please precise what Belpaoviridae is and why it is used here as an outgroup.

P5 lane 148: "All inactive retrovirus and retroelement insertions are located in the recombination-poor pericentromeric heterochromatin..." Please add reference(s) for the statement that pericentromeric regions are recombination-poor.

Figure 3A, right cartoon, beige rectangle, it is written: non-infectious retroelements (all expressed in germline cells). The statement "all expressed in germline cells" should be removed or precised, because those elements are indeed expressed but visible only in GLKD condition (not WT), this could be confusing for the readers (as it is not stated for the infectious retroviruses). Corresponding legend could be adapted as well.

Even if well referenced, please specify what GD KD and sh KD lines provide (small interfering RNA through dsRNA production and Dicer2 cleavage or small hairpin Dicer1 dependant ?).

- any additional non-essential suggestions for improving the study (which will be at the author's/editor's discretion)

For the general knowledge on this field, in particular on mobile element control (TE and ERV) in soma and germline, it would be very interesting if, in the discussion part only (no additional experiments requested here), the author could comment and give their opinion on siRNA potential involvement on their findings and if they think any connection exist between the fast evolving Ago2 protein and its involvement in TE and RNA virus control?

Point by Point Response to Reviewer's comments

Referee #1:

This is a remarkable paper providing evidence that insect endogenous retroviruses (iENV, like gypsy and ZAM), which are expressed in somatic cells (follicle cells of the ovary), gave rise to germline-expressed non-infective retroelements (like McClintock, accord and burdock). The authors show that gypsy and ZAM-infected follicle cells produce virus-like particles that infect neighboring oocytes where they integrate into the germline. In addition to establishing the perfect correlation between the presence and expression in iENVs of env-F, the env gene in the gypsy-class elements studied here, and the loss of env-F expression in retroelements, the authors explore the piRNA regulation of elements across this spectrum. Somatic piRNA expression from the flamenco piRNA cluster appears to control most of the retroviruses, but there are no 17.6 piRNAs produced from this cluster, leading to the discovery that the 77B piRNA cluster is also unidirectional and produces piRNAs against 17.6. They go on to show that 77B is necessary for control of 17.6. For me, the most impressive part of the paper centers on the analysis of rover elements. This family of elements spans the range from essentially true retroviruses (with somatic expression, viral particle formation, and other endogenous retroviral properties) to germline-restricted, non-viral particle forming transposable elements. Most impressively, they also document recombination events both within and between the retroviral and the retroelement rover LTRs. In addition to documenting all these features of rover elements, they also demonstrate the host regulation of each class, and they do molecular evolution analysis to show that the transposons are derived from the endogenous retroviruses (complete with a plausible inference of their age).

The authors make solid use of many tools from computational evolutionary genomic analysis, starting with phylogenetics. The phylogenetic tree of the Metaviridae forms the central focus of the analysis, as it contains the five major clades of the gypsy-class elements, all with a gag and pol gene. Seventeen of the lineages on this tree have env-F, and all are in the gypsy-class elements. Eleven lineages lack functional env-F and show instead varying degrees of sequence divergence, reflecting different ages of loss of function. All inactive element insertions of both the retrovirus and retroelement classes are located in pericentromeric heterochromatin, and the high LTR sequence divergence reflects their great age. Overall, the data are consistent with these iERVs being monophyletic within *D. melanogaster*, an interesting finding as it means that the diversification of this element family, spanning both endogenous retroviruses and germline-limited retroelements (transposons), occurred entirely within this species.

The experimental manipulations include RNAi knockdowns of piRNA pathway components, driven with highly tissue-specific drivers. For example, tj-Gal4-driven Zucchini shRNA accomplished knockdown of ovarian somatic piRNAs, and MTD-Gal4 driven Aubergine shRNA accomplished knockdown of piRNA expression in the ovarian germline. Single-molecule FISH provided clear evidence of changes in tissue-specific expression (e.g. that McClintock transcripts were only detected in nurse cells). The Supplementary figures show clearly the extensive differentiation in tissue specificity of the different gypsy-class elements, including targeting of the 8 different somatic cell types in the fly ovary. In particular "retroviruses with strong derepression at the RNA-seq level (ZAM, gypsy, springer, gypsy6, HMS-Beagle2)

were expressed in broad domains, while those with moderate derepression (idefix, 17.6, quasimodo, 297, gypsy5) exhibited more restricted expression niches." Moreover, viruses from the idfix subclade were predominantly expressed during early oogenesis, whereas members of the ZAM, springer and Beagle subclades were active mainly at later stages. This suggested that different iERV subclades target the germline genome at distinct stages of oogenesis. These observations suggest that infectious iERV lineages collectively occupy the full spectrum of ovarian somatic cell types, quite convincingly support the notion that they occupy different "niches" in the host.

I conclude with a simple, direct statement - this is the best paper I have read all year!

We sincerely thank the reviewer for their very encouraging, supportive, and thoughtful feedback – this is truly appreciated.

Small points, all to be considered optional:

1. I notice in the phylogenetic tree of Fig 1 that *tirant* is fairly closely related to ZAM, but there is no further mention of *tyrant*. Given the interest in its recent invasion, I was hoping the authors could say whether *tirant* the host had developed similar regulation of *tyrant* as is seen for ZAM.

This is an interesting comment. Given the recent invasion of *tirant* in *D. melanogaster* via horizontal transfer, the *tirant* retrovirus is not present in most laboratory strains and can therefore not be studied reliably with the tools used in our study. We have made an explicit comment about this in the revised manuscript (lines 176-178). In an unpublished and ongoing project, we have observed that *tirant* expression and its suppression by the host is highly similar to the related ZAM retrovirus.

2. In the reference list, Carmell et al.(2007) is listed as "in press"

This is corrected in the revised manuscript.

3. I am a bit surprised that there is no mention of siRNAs, as others have implicated them in regulation of endogenous retroviruses and somatic transposable elements.

We agree that this is a point worth highlighting. We added a brief paragraph in the Discussion section of the revised manuscript (lines 511-515).

4. The term "niche partitioning" is presented with the implication that there is some sort of competitive exclusion keeping elements from having overlapping niches. The data clearly show the conspicuous level of niche specialization, but there really is not direct evidence of competitive exclusion. This would require showing that the presence of element A in a tissue impedes the success of element B in the same tissue, either by reducing transposition rate, or by reducing host survival or fecundity. It is a minor point, but I believe that "niche specialization" is a bit more accurate here. Of course it is also a really interesting speculation that the niche specialization came about as a result of active competitive exclusion.

We agree with the suggested, more careful wording and have incorporated it throughout the revised manuscript.

Referee #2:

In this study, the authors systematically analyzed expression of endogenous retroviruses (ERVs) in the *Drosophila* ovary, comparing elements coding an envelope gene (retroviruses) and those that do not (retroelements). They find that the ancestor of the monophyletic group of ERVs likely encoded an envelope gene, which had been lost in certain lineages. They report an intriguing correlation between the presence of the envelope (and hence infectivity) and specific expression in soma versus germline: elements containing an envelope are expressed in the soma (and suppressed by the somatic piRNA pathway), whereas retroelements without an envelope are expressed in the germline (and suppressed by the germline specific piRNA pathway). Strikingly, different retroviruses are expressed in different cell types of the ovary, a process the authors refer to as niche partitioning. Using lacZ reporters, they demonstrate that these differences in expression are driven by (adaptive) changes in the LTR and 5' untranslated regions of the elements. Finally, they report an intriguing case of an ERV lineage (rover) that seems to be in a transitional state from infectious retrovirus to a retroelement.

Overall, the paper provides interesting new insights about transposon evolution, with likely relevance also beyond the *Drosophila* model systems. I found the idea of niche partitioning especially intriguing. The work is very well executed, a pleasure to read, and extensively supported by 24 (!) supplementary figures. I have only minor textual comments and suggestions for consideration.

We are grateful for the reviewer's thoughtful and encouraging feedback.

Panel indicators in Figure 1 do not correspond to the figure.

This is corrected in the revised manuscript.

Figure 1C: The difference between the red boxes indicating env gene in consensus (full-length according to the legend) and the white box under the env gene (intact ORF) for some elements is not clear to me. Please clarify.

The two data points represent the presence or absence of an intact *env* gene either in the reconstructed transposon consensus sequence (red boxes) or in the *D. melanogaster* dm6 genome. For instance, *gypsy4* and *gypsy10* are found in the dm6 genome only as fragmented copies lacking an intact *env* open reading frame. To classify their identity as retroviruses or retroelements, we reconstructed their consensus sequences based on the fragments present in the genome. This analysis indicates that both elements are retroviruses, although all insertions in dm6 are non-functional, broken copies. As the red boxes are redundant with the red colour of the respective lineage in the tree, we deleted the red boxes to avoid confusion.

Figure 4A: I did not quite understand the explanation of circle size in the legend of panel A. In the figure itself I propose to change similarity with identity.

We have revised the figure legend to explain this better. Similarity is exchanged to identity in the figure.

Figure 6D: the labeling top1, top2, top3 was not very intuitive to me. Consider rephrasing. Font of x-axis labelling is unreadable.

We revised the figure legend to make it more accessible and removed the confusing legend from the figure.

Method, piRNA cluster silencing potential: "piRNA clusters were parsed into all possible 25mer sequences". Was this done with a 1-nt offset or were the sequences parsed head-to-tail?

This was done with a 1nt offset. We mention this now in the relevant methods section.

Supplementary figure 3: delete "from" from the caption title.

This has been corrected.

Supplementary figure 5: please define triangle and asterisks.

This is now defined in the legend for this figure.

Supplementary figure 7B: the y-axis scale is missing from the right panel. Also labelling is missing for the x-axis.

This has been corrected.

Supplementary figure 10: panels of the specific knockdowns are also presented in the main figure; I wonder whether the controls are not better presented in the main figure as well. If the authors choose not to, then I suggest at least indicating in the legend that panels are duplicated (likewise for other supplementary figures in which panels are also presented in main text-figures).

We now indicated the panel duplication in the legend.

Supplementary figure 22, legend: it was unclear to me what "references" was referring to.

This has been corrected in the corresponding figure legend.

Line 110: "active and inactive lineages". How are inactive lineages defined?
this is now better explained in the methods section. (lines 589-591)

Line 183: "ZAM transcripts accumulated in yolk granules". Figure 2D suggests that the transcripts are at the periphery of the granules. The authors may want to comment on this.

We have added a brief clarification regarding this in the text (lines 191-192).

Referee #3:

- general summary and opinion about the principal significance of the study, its questions and findings

This very elegant and informative study (Senti et al., manuscript submitted: EMBOJ-2024-119776) offers insights into the co-evolution of insect endogenous retroviruses (iERVs) and their host genomes (*Drosophila*, mainly *melanogaster*), highlighting how ecological principles from evolutionary theory help to explain the complex relationship between transposable elements (TEs) and their hosts. The authors find that infectious iERVs primarily occupy somatic niches in female gonads, while non-infectious iERVs are expressed in the germline cells of the gonads. The divergent expression patterns of these iERVs are primarily driven by adaptive processes, rather than random events, contributing to their persistence within the gonad peculiar

ecosystem.

In particular, the study shows that during oogenesis, iERVs adapt to interact with the ovarian somatic and germline cells, with infectious retroviruses targeting the somatic cells and non-infectious retroelements targeting the germline cells. The key evolutionary event that enabled this shift was the acquisition of the env-F gene, which allowed retroviruses to integrate into the germline genome without directly being expressed in the germline. This event, alongside changes in CIS-regulatory sequences, fostered the diversification of iERVs, making them the most diverse Long Terminal Repeat (LTR) retrotransposon clade in species such as *D. melanogaster*.

The ovarian soma, with its greater diversity of cell types (over 7) compared to the germline (2 cell type), likely contributed to the greater divergence of iERVs compared to their germline-expressed descendants. This ecological niche partitioning may have also been driven by competition for host resources as well as viral interference. Furthermore, this study underscores the role of the piRNA genome defense system in the evolution of iERVs. Periods of unchecked iERV replication likely spurred diversification, which was later controlled by the piRNA system, with further diversification requiring a temporary loss of this control. Finally and importantly, the authors shed light on a newly discovered somatic piRNA cluster (77B) and characterise its role and importance of partial redundancy with flamenco piRNA clusters however at different "geographical" and temporal states of expression. This part is extremely elegant including the numerous functional characterization using multiple lacZ reporter systems (on ERVs and on clusters) to support the comparative sequencing analysis discoveries and hypotheses. This part also includes an elegant cluster sequence evolution comparison (including mobile element insertion) with 2 other *Drosophila* species.

This study also identifies at least 8 iERV lineages (Rover) transitioning from infectious retroviruses (iERV) to non-infectious retroelements, often associated with the loss of functional env-F. These retroelements switched from somatic to germline-specific expression through changes in CIS-regulatory sequences, potentially to avoid disrupting the germline's structure. The study highlights how recombination between retroviral and retroelement variants in natural strains of *D. melanogaster* supports the hypothesis that such recombination events drove the transition from infectious retrovirus to germline-restricted retroelement.

Broader implications of the authors findings include the evolutionary patterns shared by iERVs in insects and endogenous retroviruses in other species, such as non-insect invertebrates and vertebrates. In vertebrates, similar ecological principles and evolutionary processes may apply to ERVs, as shown by the loss of the env gene in certain ERV lineages, such as the intracisternal A particles (IAPs) in mice, which are expressed in the germline despite being targeted by the piRNA pathway. These findings suggest that mechanisms like reverse transcriptase-mediated mutagenesis and recombination may be central to the evolution of retroviruses across eukaryotic species, providing new insights into the origins and evolution of eukaryotic cis-regulatory elements.

We appreciate the positive assessment and the careful review of our work, as well as the helpful suggestions for improvement.

- specific major concerns essential to be addressed to support the conclusions

No major concern, but please check carefully a potential wrong calling (core text and legend) or labeling of the figure panels (Figure 1 mostly).

Starting from p4, lane 107: "To comprehensively characterize the envelope ORFs of Metaviridae, we considered that Env is typically translated from a spliced subgenomic transcript (Figure 1B)". Figure 1B is an alignment not the cartoon of spliced Env (also the legend is not fitting with the letter).

Same lane 118, I think a wrong figure is called: (Figure 1C; Supplementary Figure 3) check if Figure 1B has to be called here. Same lane 124: phylogenetic architectures (Figure 1D; Supplementary Figure 4A, B) not sure the right figure to call here is Figure 1D (likely Figure 1C).

We corrected the figure referencing throughout the manuscript.

Then please check before and after if the figures are called accurately, it may be a wrong labeling of the figure panels in figures provided for reviewing. Please check carefully the corresponding legend after correction as well.

We corrected the figure referencing throughout the manuscript.

- minor concerns that should be addressed

In the introduction, p3 lane 80, permissive flamenco allele is stated, it will help the less specialised readers if authors precise (and cite) what permissive (and restrictive) flamenco alleles are.

We edited the text (lines 84-90).

Lane 99 p4 please precise what Belpaoviridae is and why it is used here as an outgroup.

We added a brief explanation (lines 107).

P5 lane 148: "All inactive retrovirus and retroelement insertions are located in the recombination-poor pericentromeric heterochromatin..." Please add reference(s) for the statement that pericentromeric regions are recombination-poor.

We added a citation for this statement (line 155).

Figure 3A, right cartoon, beige rectangle, it is written: non-infectious retroelements (all expressed in germline cells). The statement "all expressed in germline cells" should be removed or precised, because those elements are indeed expressed but visible only in GLKD condition (not WT), this could be confusing for the readers (as it is not stated for the infectious retroviruses). Corresponding legend could be adapted as well.

We edited the figure and corresponding legend.

Even if well referenced, please specify what GD KD and sh KD lines provide (small interfering RNA through dsRNA production and Dicer2 cleavage or small hairpin Dicer1 dependant ?).

We explain these abbreviations in the revised manuscript (lines 169-173).

- any additional non-essential suggestions for improving the study (which will be at the author's/editor's discretion)

For the general knowledge on this field, in particular on mobile element control (TE

and ERV) in soma and germline, it would be very interesting if, in the discussion part only (no additional experiments requested here), the author could comment and give their opinion on siRNA potential involvement on their findings and if they think any connection exist between the fast evolving Ago2 protein and its involvement in TE and RNA virus control?

We added a brief paragraph about siRNAs in the Discussion section but feel that a discussion about the fast-evolving Ago2 in context of our findings is difficult due to the lack of sufficient experimental data (lines 511-515).

Dear Dr. Brennecke,

I am pleased to inform you that your manuscript has been accepted for publication in the EMBO Journal.

Yours sincerely,

Yehu Moran
Academic Editor
The EMBO Journal
